# Motor control of *Drosophila* feeding behavior

**Olivia Schwarz[1,2,3], Ali Asgar Bohra[4], Xinyu Liu[1,2], Heinrich Reichert[2], Krishnaswamy VijayRaghavan[4], Jan Pielage[1,2,3]\***

[1]Friedrich Miescher Institute for Biomedical Research, Basel, Switzerland; [2]Biozentrum University of Basel, Basel, Switzerland; [3]Division of Zoology and Neurobiology, Technical University Kaiserslautern, Kaiserslautern, Germany; [4]National Centre for Biological Sciences, Tata Institute for Fundamental Research, Bangalore, India

**Abstract** The precise coordination of body parts is essential for survival and behavior of higher organisms. While progress has been made towards the identification of central mechanisms coordinating limb movement, only limited knowledge exists regarding the generation and execution of sequential motor action patterns at the level of individual motoneurons. Here we use Drosophila proboscis extension as a model system for a reaching-like behavior. We first provide a neuroanatomical description of the motoneurons and muscles contributing to proboscis motion. Using genetic targeting in combination with artificial activation and silencing assays we identify the individual motoneurons controlling the five major sequential steps of proboscis extension and retraction. Activity-manipulations during naturally evoked proboscis extension show that orchestration of serial motoneuron activation does not rely on feed-forward mechanisms. Our data support a model in which central command circuits recruit individual motoneurons to generate task-specific proboscis extension sequences.

**\*For correspondence:** pielage@ bio.uni-kl.de

## Introduction

Locomotion and behavioral motor sequences are generated by a precise movement of selected body parts. These movements include both the coordination of individual elements of an appendage or limb to generate stereotyped serial motor patterns and bilateral interlimb coordination. In the last years, significant progress has been made towards the identification of central neuronal circuitries mediating and controlling the alternation of limb movement necessary for walking or swimming in both invertebrate and vertebrate model systems (*Berkowitz et al., 2010*; *Guertin, 2009*; *Marder et al., 2005*; *Talpalar et al., 2013*). These studies demonstrated that in many cases local central pattern generators (CPGs) and reciprocal inhibitory interneuron networks coordinate the alternating activation of limb motor units (*Berkowitz et al., 2010*; *Borgmann and Büschges, 2015*; *Büschges et al., 2011*; *Crone et al., 2008*; *Guertin, 2009*; *Lanuza et al., 2004*; *Marder et al., 2005*; *Talpalar et al., 2013*). Similarly, CPGs are involved in the generation and coordination of stereotyped motion patterns of limb or appendage segments depending on alternating extensor-flexor muscle activation (*Grillner, 2003*; *Talpalar et al., 2011*; *Tripodi et al., 2011*; *Zhang et al., 2014*). Intra-limb coordination of body parts has been mainly explored using vertebrate limb movement, turtle scratch behavior and directed locomotion of locust legs (*Berkowitz and Laurent, 1996*; *Calas-List et al., 2014*; *Durr and Matheson, 2003*; *Guzulaitis et al., 2014*; *Machado et al., 2015*; *Snyder and Rubin, 2015*; *Stein, 2010*). In addition, analysis of Drosophila larval locomotion recently provided insights into the generation of temporally delayed but overlapping muscle activation patterns (*Zwart et al., 2016*). This study demonstrated similar segregation of premotor excitatory input

as observed in vertebrates (*Bikoff et al., 2016*; *Goetz et al., 2015*; *Tripodi et al., 2011*) and showed that inhibitory interneuron input mediates phase delay of intrasegmental motoneuron (MN) activation (*Zwart et al., 2016*).

Despite these advances, for complex reaching-like behaviors we currently have only a limited understanding regarding the circuit architecture that controls individual MN activation to elicit and coordinate these precise temporal and spatial motion patterns.

Here, we use the stereotypic motor response of *Drosophila melanogaster* proboscis extension to address in vivo the cellular and circuit mechanisms underlying the serial activation pattern of muscle groups necessary to coordinate a reaching-like behavior. The proboscis extension response (PER) is part of the sensory-motor taste circuitry of adult Drosophila (*McKellar, 2016*). The proboscis is the feeding organ of flies and is used for both taste cue detection and food ingestion (*Dethier, 1976*; *Masek and Scott, 2010*; *Shiraiwa and Carlson, 2007*; *Wang et al., 2004*). Comparable to mammals, gustation in flies is based on a limited number of modalities which are perceived by gustatory receptor neurons (GRNs) present in taste sensilla on the proboscis, legs, wings, and ovipositor. Stimulation with an attractive stimulus (sweet) will trigger the extension of the proboscis towards the food source while aversive stimuli (bitter) will prevent the PER (*Clyne et al., 2000*; *Dunipace et al., 2001*; *Falk et al., 1976*; *Hiroi et al., 2004*; *Montell, 2009*; *Scott et al., 2001*; *Singh, 1997*; *Stocker, 1994*; *Thorne et al., 2004*; *Yarmolinsky et al., 2009*).

For a number of reasons Drosophila proboscis extension represents an ideal model system to unravel the structural and functional basis of a serial motor action. First, the PER represents an innate, sequential behavior that can be subdivided into a discrete number of movement steps (*Flood et al., 2013*). This motor sequence likely requires activation of different muscle groups at distinct time points within the PER sequence, implying a precise temporal orchestration of upstream MN activity. Second, the PER can reliably and noninvasively be elicited in living flies simply by applying a positive gustatory stimulus to GRNs (*Shiraiwa and Carlson, 2007*). Third, the MNs innervating the proboscis reside in a specific, highly regionalized brain region, the subesophageal zone (SEZ, nomenclature according to *Ito et al., 2014*) (*Hampel et al., 2011*; *Rajashekhar and Singh, 1994*). It is thought that the relay of gustatory sensory information from GRNs to MNs occurs mainly within the SEZ (*Altman and Kien, 1987*; *Dunipace et al., 2001*; *Stocker, 1994*; *Thorne et al., 2004*; *Wang et al., 2004*).

Importantly, stereotypic proboscis extension is also part of additional innate behavioral programs. The proboscis is partially extended both during fly grooming to enable cleaning of the proboscis (*Hampel et al., 2015*; *Seeds et al., 2014*) and during the male courtship to enable contact to the female fly (courtship licking) (*Hall, 1994*; *Nichols et al., 2012*). As these movements differ significantly from each other at least three independent motor programs controlling proboscis extension must exist.

The current description of the Drosophila proboscis motor system largely relies on comparative anatomical studies of the proboscis musculature based on cross-sections of the adult head in different fly species (*Graham-Smith, 1930*; *Miller, 1950*). First insights regarding the anatomy of MNs were obtained using backfilling studies (*Rajashekhar and Singh, 1994*) and by selective expression of marker genes in MNs innervating the musculature of the pharyngeal pump (*Tissot et al., 1998*). More recently, by gaining genetic access to individual MNs a functional analysis enabled the characterization of the role of a single MN during feeding induced proboscis extension (*Gordon and Scott, 2009*) and of MNs contributing to food intake by controlling pharyngeal contractions (*Manzo et al., 2012*; *Tissot et al., 1998*). However, to gain insights into the principles underlying the motor program controlling proboscis movement a comprehensive neuroanatomical and functional characterization of proboscis muscles and MNs is essential.

Here, we first analyze the sequential features of the motion pattern underlying the PER and provide a comprehensive morphological description of proboscis MNs and muscles. Using a MARCM (Mosaic Analysis with a Repressible Cell Marker) approach (*Lee and Luo, 1999*) we identify and characterize cell body position, dendritic arborization, nerve projections, and muscle innervation patterns of all proboscis MNs at the single cell level. Using a functional behavioral screen, we then identify essential MNs controlling the serial motor sequence of the PER. Light and temperature-mediated activation and silencing of genetically identified MNs in vivo enables us to assign individual MNs to all major steps of the motor sequence controlling proboscis extension and retraction. Finally, by using targeted neuronal activity manipulations during natural, stimulus-evoked PER we demonstrate

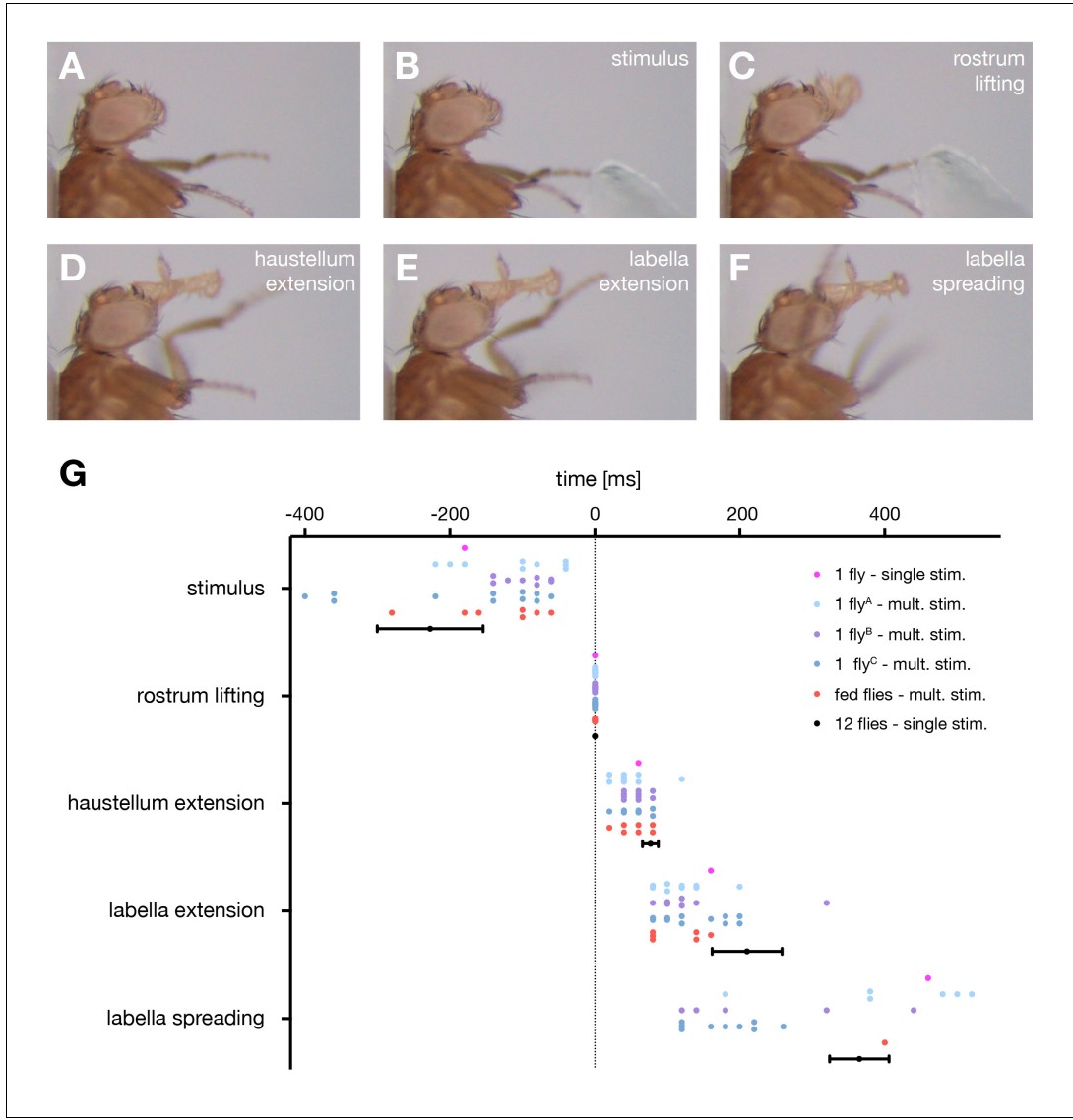

**Figure 1.** The motor sequence of the proboscis extension response. (A–E) In response to sucrose stimulation to the leg the proboscis is extended in a stereotypic motion pattern: (A) fly before stimulus, (B) sucrose stimulation, (C) rostrum lifting, (D) haustellum extension, (E) labella extension, (F) labella spreading. (G) Temporal quantification of proboscis extensions. The initiation time point of each step was determined in the video sequence and plotted with rostrum lifting set to zero. Data are shown for: single fly (magenta), multiple stimulations of three individual flies (A n = 13, B n = 10, C n = 13 stimulations), fed flies (red, n = 4 animals, 7 stimulations), mean ± SEM of the second stimulation of 12 individual flies (black). The following data points are not displayed on the graph: FlyA, stimulus (−28), labella spreading (+35); flyB, stimulus (−31, −54); flyC: labella spreading (32, 46). Statistical comparison of flies A–C (Mann-Whitney U test) revealed small significant differences for the following data points: A vs C, labella extension – labella spreading, p=0.0006; B vs C, stimulus – rostrum lifting, p=0.0304). No significant differences were detected when comparing fed flies to individual flies or to the 12 control flies. However, these flies failed to spread the labella in response to leg stimulation but not when stimulated at the labella. See also *Video 1*.

that the motor sequence units act independently from each other. Our study indicates that the serial PER action sequence is centrally programmed and does not represent a chain reflex sequence.

## Results

### Characterization of the PER motor sequence

First, we aimed to determine the precise motion pattern underlying the PER motor program. Therefore, we monitored and quantified proboscis movements in 14 starved and immobilized wild type ($w^{1118}$) flies in response to sucrose stimulations of the anterior legs. Our analysis revealed that the PER program consists of four major extension steps prior to food ingestion. Upon sucrose stimulation flies (1) lift the rostrum, (2) extend the haustellum, (3) extend the labella and (4) spread the labella to prepare for food intake (*Figure 1A–F*, *Video 1* – see slow motion). This sequence is consistent with the reported sequences both during natural feeding and sucrose stimulation (*Dethier, 1976*; *Flood et al., 2013*; *Gordon and Scott, 2009*) with the exception of the labella extension step that has not been described before. Importantly, this sequence was highly stereotypic both within individual flies and across multiple flies (*Figure 1G*). We observed a deviation from this sequence only in 4 out of 93 stimulations (n = 14 flies) in which labella extensions preceded haustellum extensions. Between individual flies small alterations in the temporal profiles of individual movement steps could be observed (*Figure 1G*). These alterations are likely not a consequence of the feeding status of the flies as we did not observe significant deviations of the temporal sequence in fed flies compared to starved flies (*Figure 1G*).

### Identification of proboscis musculature

We next aimed to unambiguously identify all muscle groups potentially contributing to proboscis movement and food ingestion. We used a muscle specific reporter (MHC-GFP; *Chen and Olson, 2001*) to visualize the position and organization of all muscles within an intact head capsule and proboscis (*Figure 2A,B*). Our whole-head preparation allowed us to visualize all muscles in their natural position and enabled identification of muscle groups that were not recognized as distinct groups in prior studies (*Miller, 1950*; *Rajashekhar and Singh, 1994*). For nomenclature, we follow the numbered system introduced by *Miller (1950)*. The analysis of muscle organization at different focal positions within the head capsule resulted in a number of novel findings. Muscle 1 represents the largest muscle group extending through the entire head capsule (*Figure 2A,B*). Analysis of the flanking muscles revealed that muscle 2 is comprised of two independent muscles with unique attachment sites and different expression levels of the MHC-GFP reporter (*Figure 2B*). Similarly, the in situ visualization of muscle groups surrounding the pharynx revealed novel aspects of muscle group organization (*Figure 2b'*, displayed at higher exposure levels). As previously described, muscle group 12 is composed of two different muscles, 12–1 and 12–2 (*Flood et al., 2013*; *Figure 2b'*). In addition, our data shows that the large muscle group 11 can be subdivided into three distinct muscle sets (11–1, 11–2, and 11–3) that attach to the upper sclerotized plate of the pharyngeal pump at unique angles (*Figure 2b'*). Within the haustellum muscles 6 and 7 share posterior attachment positions but connect to the dorsal and ventral part at the anterior end of the haustellum, respectively (*Figure 2b''*). Muscle 8 forms a connection between the dorsal and ventral parts of the labella, orthogonal to muscles 6 and 7 (*Figure 2b''*). Based on these data, the proboscis musculature consists of 17 individual muscles forming 13 major muscle groups.

### Proboscis motoneurons are located in the subesophageal zone

To characterize the MNs innervating these muscle groups we first visualized MN cell bodies by backfilling the proboscis nerves (labial and pharyngeal nerve) with rhodamine-labeled dextran dye. These experiments, recapturing an original analysis of *Rajashekhar and Singh (1994)*, revealed 20 pairs of bilaterally symmetric MNs, with MN somata arranged in two bilaterally

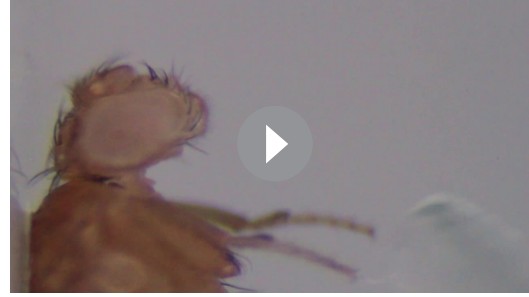

**Video 1.** Proboscis extension sequence in wild type flies (related to *Figure 1*). This video shows a side view sequence of a sucrose-evoked proboscis extension of a wildtype ($w^{1118}$) fly first in real time followed by slow motion (0.1 x speed).

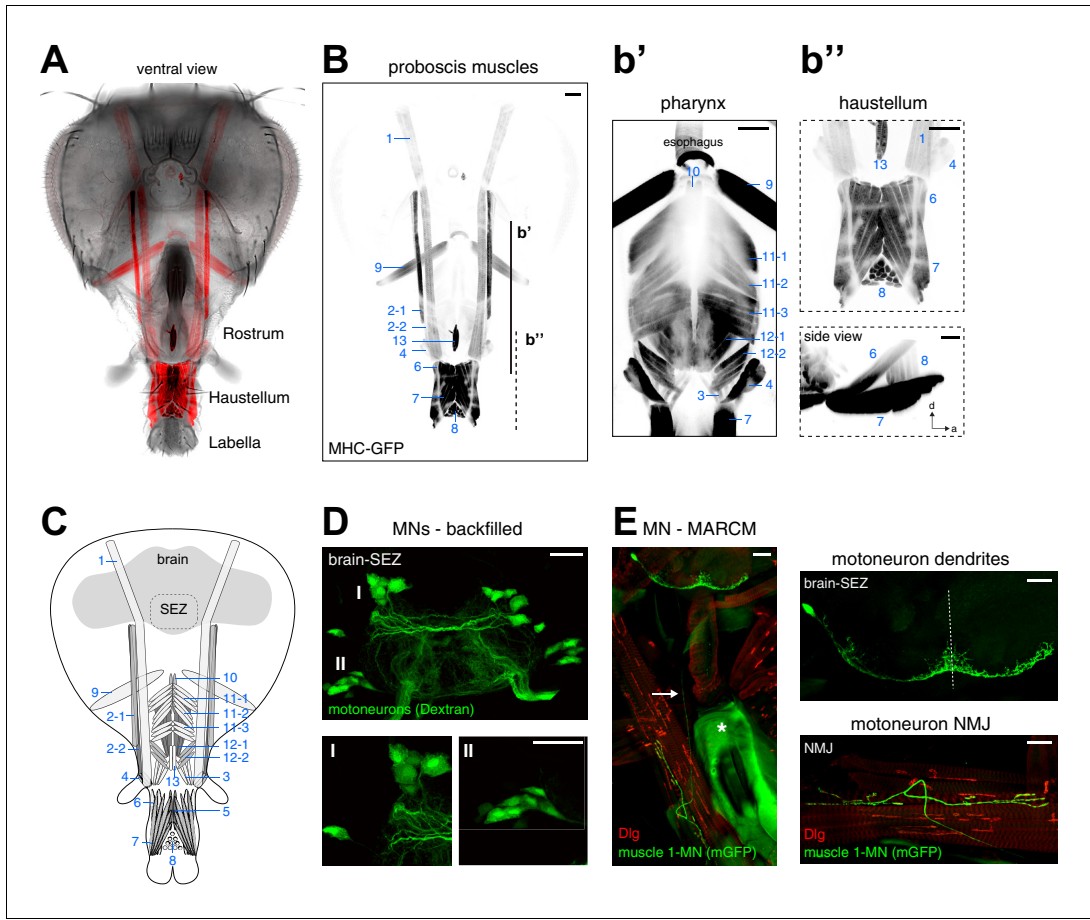

**Figure 2.** Muscles and motoneurons of the Drosophila proboscis. (**A**) Whole mount preparation of a Drosophila head. Muscle-specific expression of GFP (MHC-GFP) reveals the position of all proboscis muscles (red) in the head (bright field image). (**B**) Same head as in (**A**) with muscles displayed in black (inverted). Pharynx and haustellum muscles are shown with adjusted settings and individually in b' and b''. Blue numbers indicate individual muscle groups. Scale bars, 50 μm. (**C**) Schematic drawing of head muscles. (**D**) Proboscis MNs. Upper panel, backfilling of all MN axons innervating the proboscis musculature reveals MN cell bodies within the SEZ. Two clusters of MN cell bodies (DMA, dorsal medial anterior; VLP, ventral lateral posterior) are present on both sides of the midline. Lower panel, enlargements of the DMA (I) and VLP (II) clusters. Scale bars, 20 μm. (**E**) Single cell MARCM clone of a proboscis MN innervating muscle 1. Left, overview showing MN cell body and dendritic arborization in the brain that is connected by a single axon (arrow) to the NMJ on muscle 1. The asterisk indicates the pharyngeal plate. Right, enlargements of the SEZ (top) and NMJ (bottom). In all panels, the MN is marked by the expression of mCD8-GFP (green) and postsynaptic sites are labeled using anti-Discs large (Dlg, red). Scale bars, 20 μm.

symmetric clusters in the SEZ (*Figure 2D*). Consistent with prior reports we observe that all dendritic MN arborizations are confined to the SEZ. Thus, at least 20 distinct MNs in each brain hemisphere control the activation of the 13 muscle groups contributing to either proboscis movement or pharynx-mediated food uptake.

## Developmental origin and neuroanatomy of proboscis MNs

To characterize the neuroanatomical features of all MNs in detail and to gain insights into their developmental origin we used the MARCM technique that allows genetic labeling of individual MNs at the time of their birth (*Lee and Luo, 1999*). Proboscis MNs can be visualized using the Gal4 driver line OK371 that labels all glutamatergic neurons (*Daniels et al., 2008*). Interestingly, we only recovered MN clones when heat-shock mediated flippase activation was induced during early embryogenesis (0–12 hr after egg-laying (AEL)) but not when the activation was performed during late

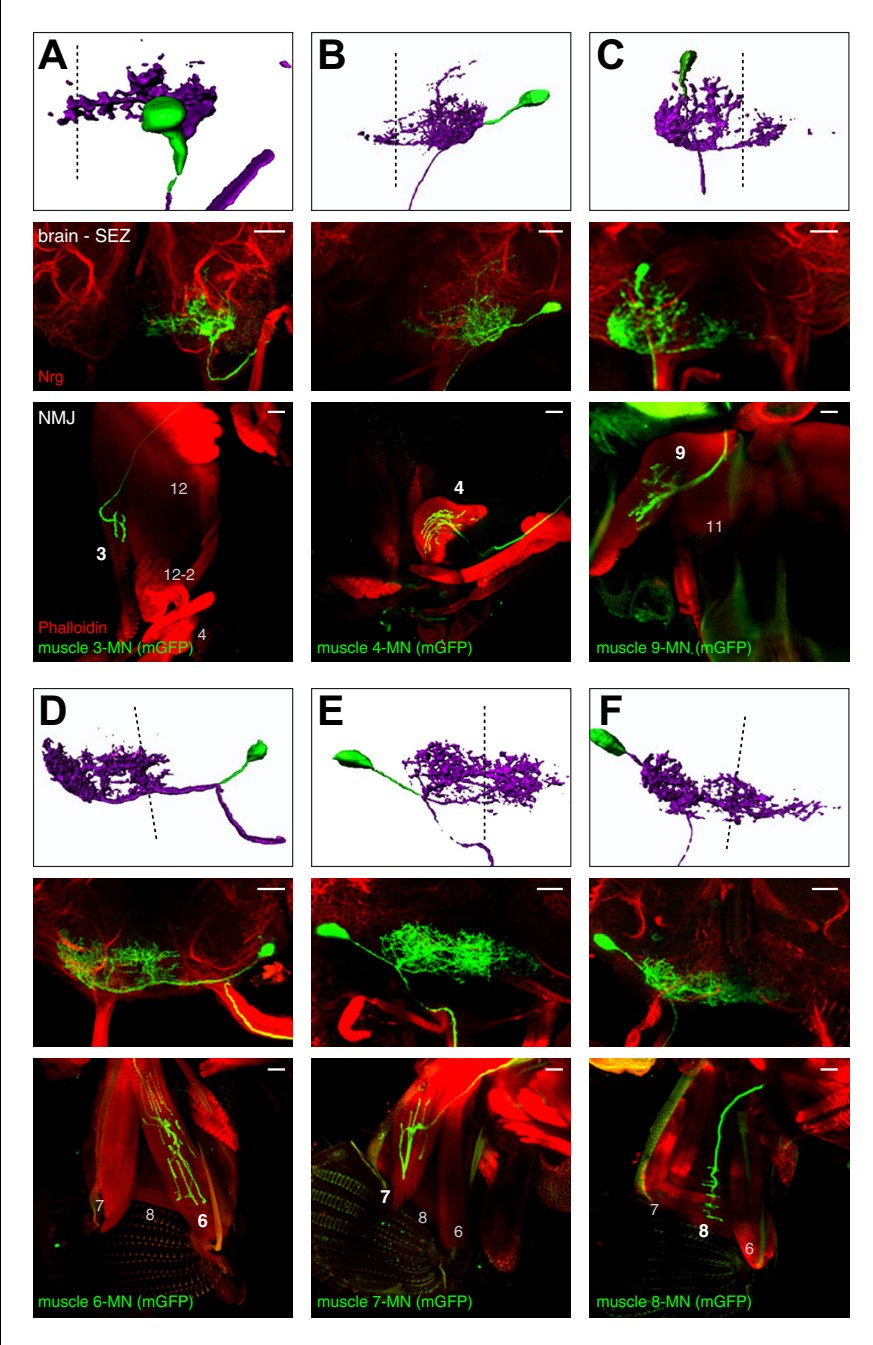

**Figure 3.** Morphology of MNs contributing to proboscis motion. Single cell MARCM clones of MNs innervating muscle 3 (**A**), muscle 4 (**B**), muscle 9 (**C**), muscle 6 (**D**), muscle 7 (**E**), and muscle 8 (**F**) are shown. For each MN type, brain localization (middle panels) and muscle innervation (lower panels) are shown. In all panels, MNs are marked by the expression of mCD8-GFP (green), the neuropil is visualized using an anti-Neurotactin antibody (Nrt, red, middle panels), and muscles are labeled using rhodamine-conjugated phalloidin (red, bottom row). A digital reconstruction of each MN is shown (top panel, dotted line indicates midline). Cell bodies are artificially colored in green and neurites in magenta. Scale bars, 20 μm.

embryogenesis or during larval stages. Furthermore, we never recovered multiple MNs within a brain hemisphere (*Figures 3* and *4*). This is in contrast to the development of leg MNs that occurs throughout larval development and is coupled to the development of the adult leg with individual

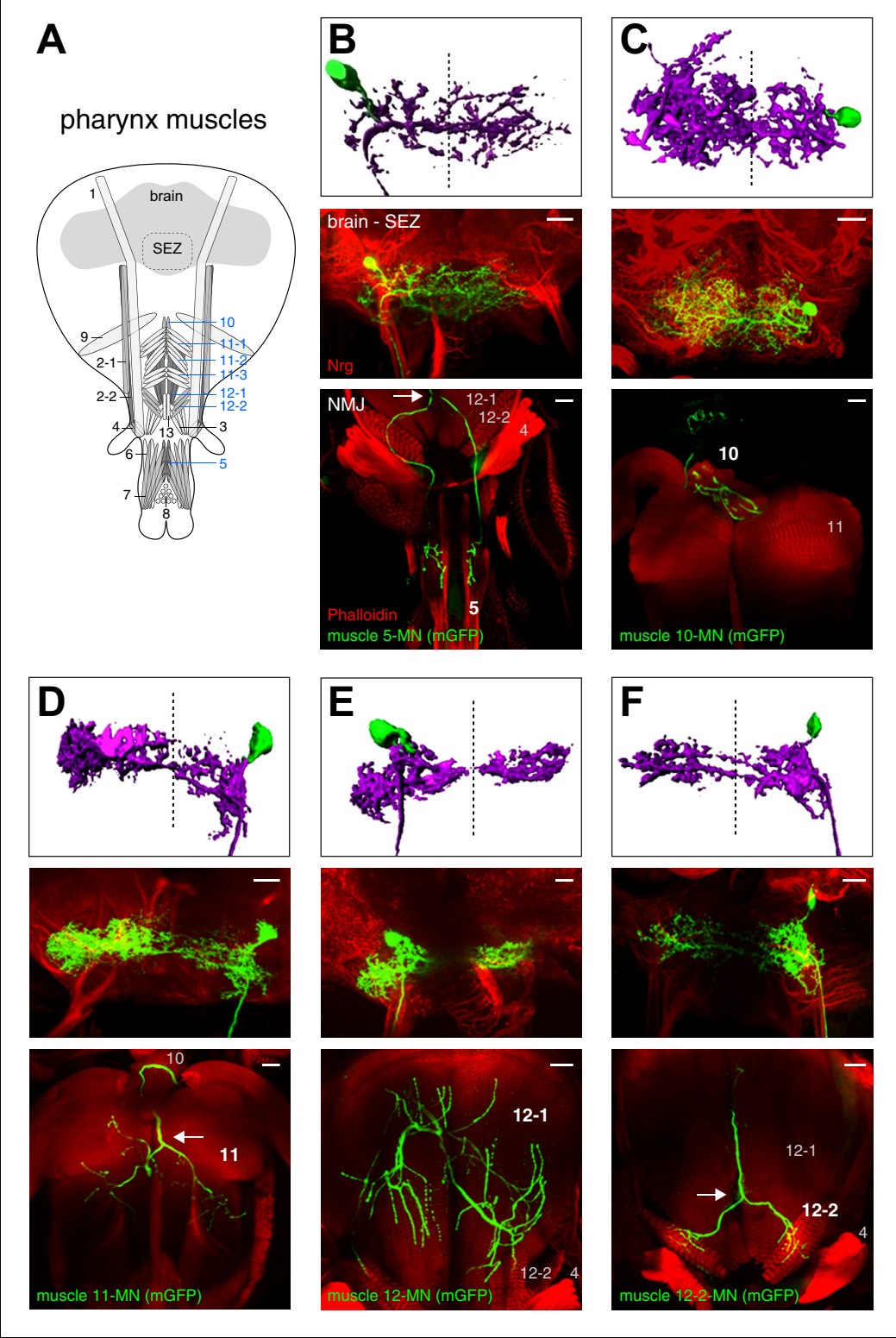

**Figure 4.** Morphology of MNs contributing to food ingestion. (**A**) Schematic drawing of the muscles in the fly head. Muscles implicated in food ingestion are marked in blue. Single cell MARCM clones of MNs innervating muscle 5 (**B**), muscle 10 (**C**), muscle 11 (**D**), muscle 12 (**E**), and muscle 12–2 (**F**) are shown. For each MN type (**B**–**F**), brain localization (middle panels) and muscle innervation (bottom panels) are shown. In all panels, MNs are marked by the expression of mCD8-GFP (green), the neuropil is visualized using Nrt (red, middle row), and

*Figure 4 continued on next page*

*Figure 4 continued*

muscles are labeled using rhodamine-conjugated phalloidin (red, bottom row). A digital reconstruction of each MN is shown (top panel, dotted line indicates midline). Cell bodies are artificially colored in green and neurites in magenta. Scale bars, 20 μm.

neuroblasts giving rise to a large number of MNs (*Baek and Mann, 2009*; *Brierley et al., 2012*). We analyzed the dendritic arborization, axon projection and muscle innervation pattern of 96 proboscis MN clones (*Figures 2E*, *3* and *4*) and defined MN types as MNs innervating the same muscle group. A minimum of two independent single cell clones was obtained for each MN type (*Figures 3* and *4* and data not shown) with the exception of MNs innervating muscle 13 (no clones recovered). Based on their innervation patterns, the twelve analyzed MN types can be subdivided into two major groups. Eight MN types innervate target muscles only on the ipsilateral side of the proboscis (with respect to the soma); these MNs innervate muscles that are involved in the extension, retraction, and positioning of the proboscis and mouthparts (this paper; *Figures 2E* and *3*, and below). Strikingly, the axons of the remaining four MN types bifurcate and simultaneously innervate bilateral symmetric target muscles associated with the pharyngeal pump (*Figure 4*). These muscle groups are thought to mainly control food ingestion and pumping (*Manzo et al., 2012*; *Tissot et al., 1998*).

## MNs innervating muscles controlling proboscis movement

The eight muscle groups innervated by ipsilateral MN types have been previously categorized based on anatomical criteria in Drosophila by *Miller (1950)* and in the blowfly by *Graham-Smith (1930)*. Functional data thus far only exists for muscle 9 that has been demonstrated to control rostrum lifting (*Gordon and Scott, 2009*). Representative single cell clones of MN types that innervate seven of these muscles are shown in *Figure 2E* and *Figure 3*. As the precise role of these muscles for proboscis movement has not yet been established through functional analysis we utilize the target muscle number and not the anatomically based role for MN classification throughout this manuscript. The MN innervating muscle 1 (= MN1) has both ipsilateral and contralateral dendritic arborizations and innervates the ipsilateral muscle through the labial nerve (*Figure 2E*). Based on the co-staining with the postsynaptic muscle marker Discslarge (Dlg) it is evident that at least one additional MN innervates muscle 1. In general, MNs controlling proboscis movement differ significantly in the localization and complexity of dendritic arborization. MNs 3 and 4 display almost exclusively ipsilateral dendritic arborization (*Figure 3A,B*), while MNs 9 and 8 have predominantly ipsilateral arborization with minor extensions to the contralateral side (*Figure 3C,F*). In contrast, MN7 has similar dendritic arborizations on both sides (*Figure 3E*) while MN6 displays predominant contralateral arborization with only minor extensions to the ipsilateral side (*Figure 3D*). Thus, while all these MNs strictly innervate ipsilateral located muscles they receive presynaptic input either predominantly ipsilateral, equal from both sides or predominantly from the contralateral side.

## Proboscis motoneurons innervating pharyngeal muscles

Some of the muscle groups innervated by the bifurcating MNs (5, 10, 11, and 12, *Figure 4A*) have been previously associated with food ingestion and pumping (*Flood et al., 2013*; *Graham-Smith, 1930*; *Manzo et al., 2012*; *Miller, 1950*; *Tissot et al., 1998*). The general anatomy of these four MN types is highly stereotypic. All MN axons project through the pharyngeal nerve, bifurcate into two bilateral axon branches and innervate homologous muscles on both sides of the midline. These MNs display similar dendritic arborizations in both brain hemispheres and the innervation pattern on the two homologous bilateral muscles is almost identical (*Figure 4B–F*). Interestingly, prior analysis of the MNs innervating muscle groups 11 and 12 using specific Gal4 lines demonstrated that they have contralateral homologs (*Manzo et al., 2012*; *Tissot et al., 1998*). Indeed, in our MARCM analysis we identified single MNs innervating muscle group 11 bilaterally with the cell body present in either the left or right brain hemisphere (data not shown). Thus, pharyngeal muscles on both sides are innervated by two bilateral homologous MNs with highly overlapping dendritic arborizations.

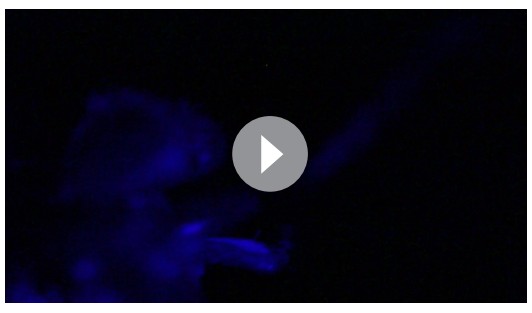

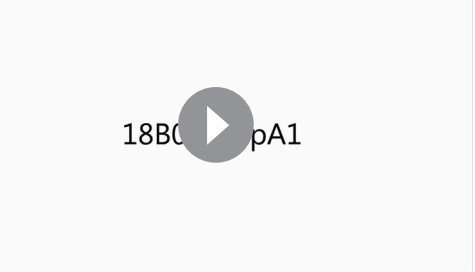

**Video 2.** Activation of Gr5a-expressing sweet sensory neurons using Chrimson. This video shows a continuous sequence of a Gr5a-Gal4>Chrimson fly at the control wavelength (475 nm), at the activation wavelength (633 nm), and at the control wavelength. First in real time followed by slow motion (0.4 x speed).

**Video 3.** Activation of GMR18B07, repoGal80 neurons using TrpA1 (related to *Figure 5*). This video shows a GMR18B07, repoGal80 > TrpA1 fly at control and activation temperatures. Order of sequence: Side view at 22°C, side view at 29°C, top view at 29°C, and side view at 22°C.

## Gal4-mediated genetic control of proboscis MNs

Next we aimed to assign functional roles to the MNs innervating proboscis musculature. Thus far, only the role of the MN innervating muscle 9 has been adequately studied by selective activation and silencing using a specific Gal4-driver line (*Gordon and Scott, 2009*). To identify the functional role of the MNs and their target muscles and to investigate the circuit mechanisms controlling proboscis extension and retraction we aimed to genetically control individual MNs. We performed a functional screen using the Gal4-UAS binary expression system (*Brand and Perrimon, 1993*) to identify Gal4-driver lines selectively expressing in different proboscis MNs. We used two publically available enhancer-Gal4 line collections (GMR-Gal4 lines, Bloomington Drosophila Stock Center, *Jenett et al., 2012*; VT-Gal4 lines, Vienna Drosophila RNAi Center, *Kvon et al., 2014*) which express the yeast transcription activator protein Gal4 in a random but fixed subset of neurons (*Pfeiffer et al., 2008, 2010*). Gal4-lines were prescreened for expression within the SEZ and then crossed to UAS-effector lines enabling either neuronal activation or silencing (*Hamada et al., 2008*; *Kitamoto, 2001*; *Klapoetke et al., 2014*). Artificial activation of Gr5a-Gal4 sweet sensory neurons by expressing either the heat-activatable Na$^+$-channel TrpA1 (*Hamada et al., 2008*) or the red-shifted Channelrhodopsin2 Chrimson (*Klapoetke et al., 2014*) resulted in repetitive complete extensions of the proboscis mimicking natural activation by sucrose (*Video 2*). In contrast, it has been reported that constant activation of MN9 caused a constant displacement of the proboscis

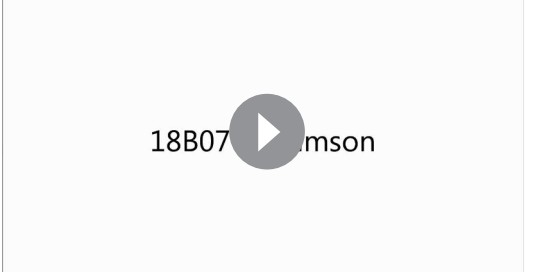

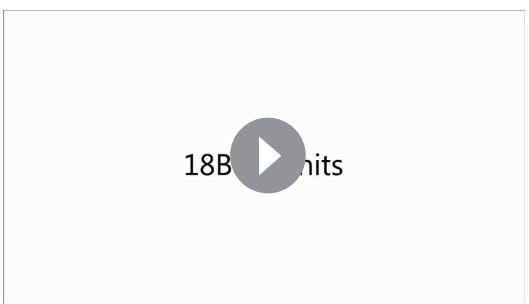

**Video 4.** Activation of GMR18B07, repoGal80 neurons using Chrimson (related to *Figure 5*). This video shows a GMR18B07, repoGal80 > Chrimson fly at the control wavelength (475 nm), then at the activation wavelength (633 nm), and at the control wavelength.

**Video 5.** Silencing of GMR18B07, repoGal80 neurons using shi$^{ts}$ (related to *Figure 5*). This video shows sucrose stimulations of a GMR18B07, repoGal80 > shi$^{ts}$ fly at 22°C, at 29°C, and at 22°C, displayed at a 0.5 x speed.

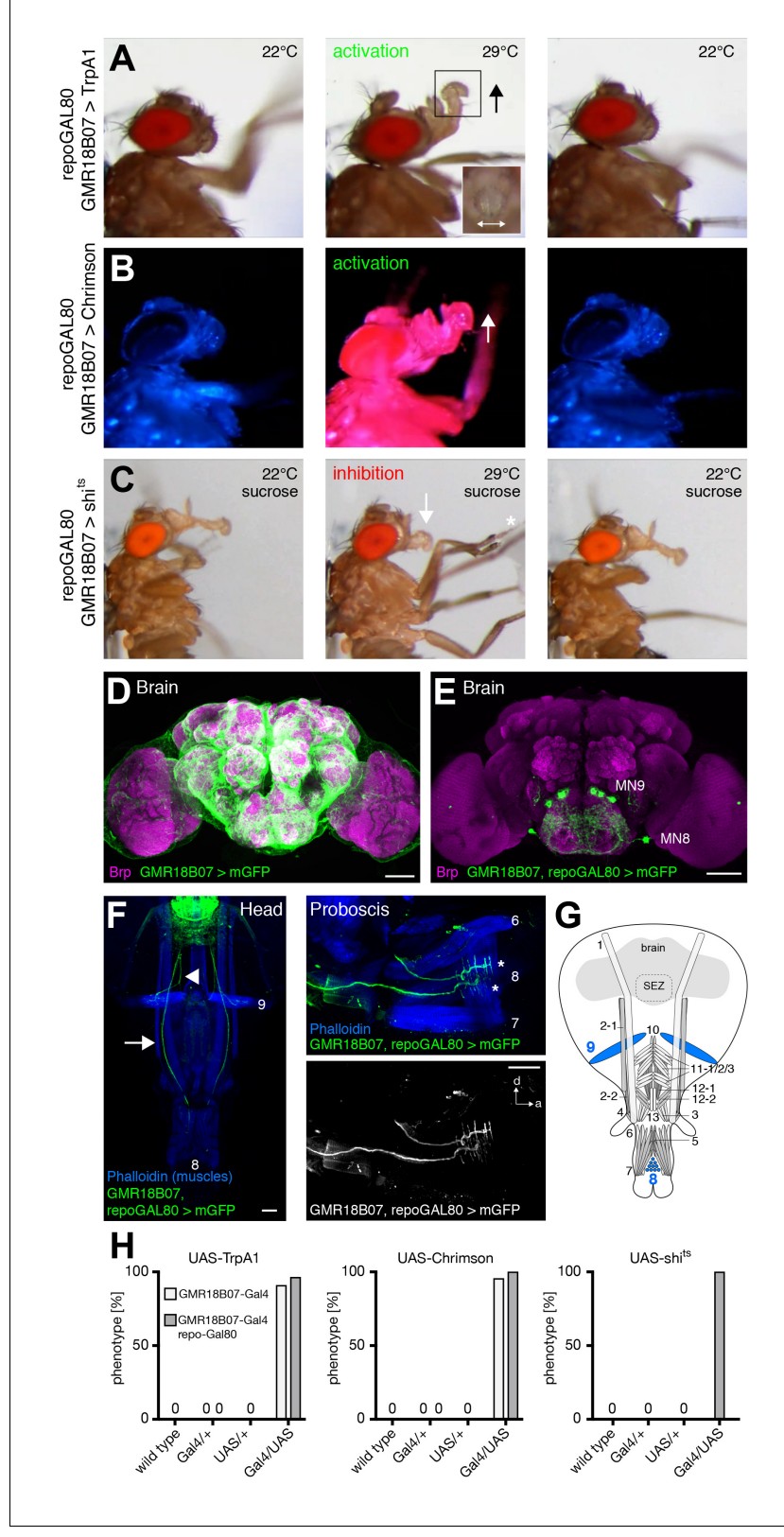

**Figure 5.** GMR18B07 neurons control rostrum lifting and labella spreading. (**A**) Artificial activation of GMR18B07, repoGal80 neurons using TrpA1. Heat induced activation elicits rostrum lifting (middle panel, arrow) and labella spreading (double arrow). The inset shows a top view of the spread labella. At the control temperature before (left panel) and after (right panel) activation the proboscis is retracted. (**B**) Artificial activation of GMR18B07, repoGal80

*Figure 5 continued on next page*

*Figure 5 continued*

neurons using Chrimson. Red light induced activation (middle panel) elicits rostrum lifting (arrow) and labella spreading. At blue light before (left panel) and after (right panel) activation the proboscis is retracted. (C) Heat induced silencing of GMR18B07, repoGal80 neurons using shibire[ts]. Flies at the permissive temperature show full PER upon 200 mM sucrose stimulation (left and right panel). At the restrictive temperature, these flies fail to lift the rostrum (middle panel, arrow) upon 200 mM sucrose stimulation (asterisk) but still extend the haustellum. (D) Expression pattern of GMR18B07 in the adult central brain. Cells are marked by the expression of mCD8-GFP (green) and the neuropil is visualized using the presynaptic active zone marker Bruchpilot (Brp, magenta). (E) Suppression of glia cell expression using repo-Gal80 reveals four pairs of bilateral neurons in the SEZ. (F) Whole head preparation of GMR18B07, repoGal80 > mCD8-GFP animals (left panel) reveals expression in two bilateral pairs of MNs (green) with one pair innervating muscle 9 (axon marked by arrowhead) and one muscle 8 (axon marked by arrow). Side view of the proboscis (right panels) shows innervation of muscle 8 (asterisks). Muscles are marked by phalloidin (blue). (G) Schematic drawing of the head muscles with innervated muscles highlighted in blue. Scale bars, 50 μm. (H) Quantification of the behavioral phenotypes in control and experimental animals. Numbers and significances are listed in *Supplementary file 1*. See also *Figure 5—figure supplement 1* and *Videos 3*, *4* and *5*.

The following figure supplement is available for figure 5:

**Figure supplement 1.** Analysis of the PER sequence during MN silencing.

consistent with the contraction of muscle 9 (*Gordon and Scott, 2009*). Based on these results, we hypothesized that constant activation of MNs should elicit a constant change of proboscis posture at the activation temperature (TrpA1) or upon red light stimulation (Chrimson).

## MNs controlling rostrum lifting and labella spreading

Artificial activation of flies expressing TrpA1 using GMR18B07-Gal4 resulted in a constant lifting of the rostrum identical to the behavioral pattern previously described for MN9 activation (E49-Gal4; *Gordon and Scott, 2009*) (*Figure 5A,H* and *Video 3*). In addition to rostrum lifting the flies also spread their labella at the activation temperature (29°C) (*Figure 5A*, inset) but not at control temperature (22°C). To confirm these results, we next used Chrimson as an alternative activation method. Upon red light stimulation, the rostrum was lifted and the labella were spread. Importantly, Chrimson-mediated activation allowed precise temporal control of the behavior as rostrum lifting correlated perfectly with red light exposure (*Figure 5B,H* and *Video 4*). To investigate whether the neurons expressing Gal4 are not only sufficient but also necessary for rostrum lifting and labella spreading we next silenced these neurons using the temperature-sensitive version of Dynamin, shibire[ts] (*Kitamoto, 2001*). At the permissive temperature (22°C) the flies were able to fully extend the proboscis towards a positive stimulus (tissue soaked in 200 mM sucrose solution). In contrast, at the restrictive temperature (29°C, please see Materials and methods for details) the flies were no longer able to lift the rostrum upon sucrose stimulation (*Figure 5C* middle panel, 5 hr and *Video 5*; GMR18B07, repo-Gal80 > shi[ts] animals, see below). Importantly, this behavior was completely reversible as shifting to the permissive temperature restored full proboscis extension upon sucrose stimulation (*Figure 5C* right panel and *Video 5*). Thus, the failure to lift the rostrum was indeed due to the acute inhibition of GMR18B07 neurons and not due to habituation or proboscis damage. In contrast to the efficient inhibition of rostrum lifting we did not observe a significant failure to spread the labella in these flies. Together, these results demonstrate that GMR18B07 neurons are both sufficient and required for rostrum lifting and at least partially involved in the control of labella spreading.

We next analyzed the expression pattern of the GMR18B07-Gal4 line using membrane-tagged GFP (UAS-mCD8-GFP) as a reporter. This analysis revealed a broad expression in glia cells throughout the brain preventing characterization of SEZ neurons (*Figure 5D*). To restrict Gal4-expression to neurons we co-expressed the Gal4 inhibitor Gal80 in all glial cells (repo-Gal80; *Awasaki et al., 2008*). Absence of glial expression revealed 4 pairs of bilaterally symmetric neurons within the SEZ (*Figure 5E*). To identify potential MNs we used the whole head preparation method that enables simultaneous analysis of the SEZ and all proboscis muscles (*Figure 5F*, see Materials and methods).

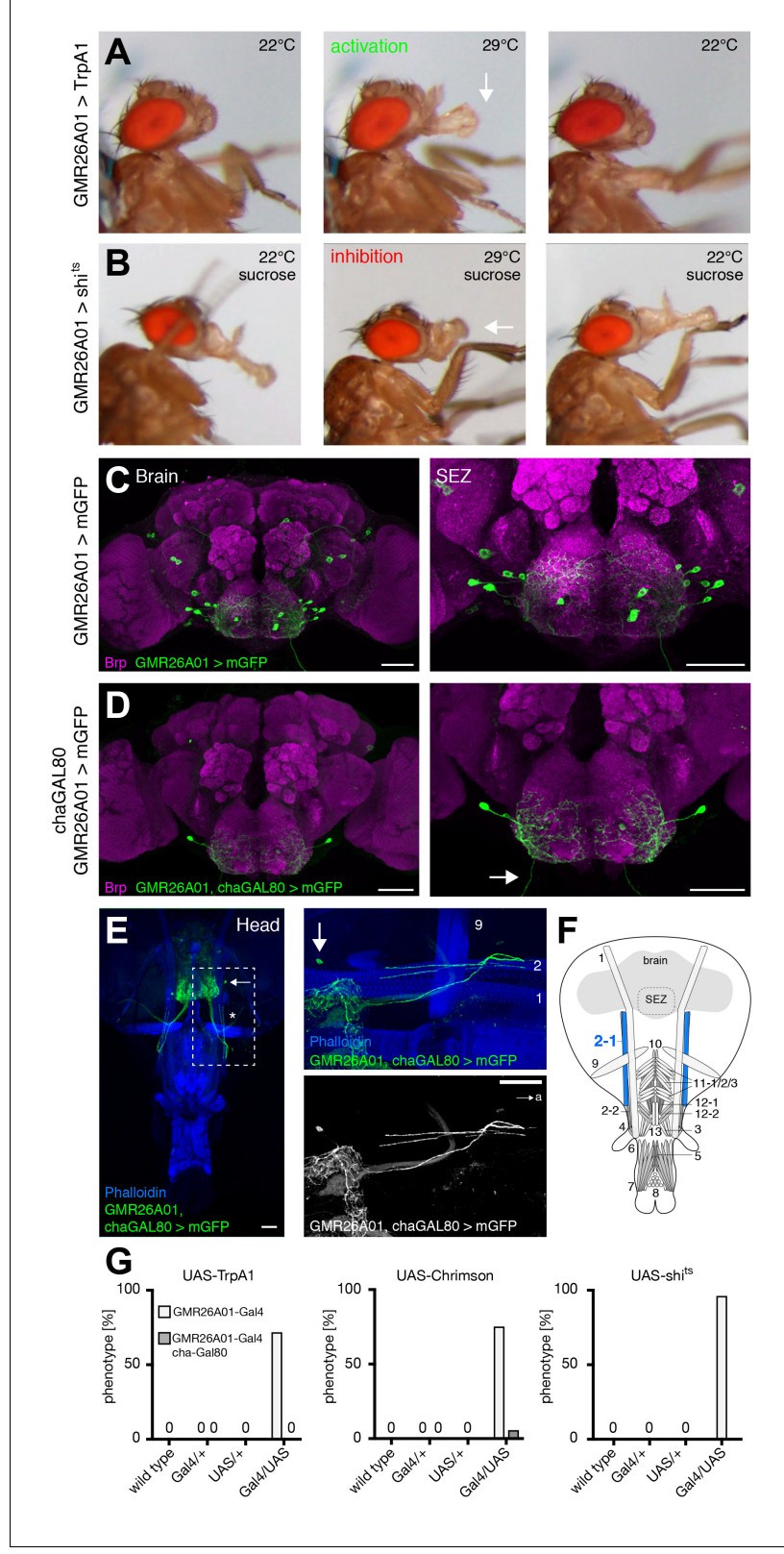

**Figure 6.** GMR26A01 neurons are sufficient and required for haustellum extension. (**A**) Artificial activation of GMR26A01 neurons using TrpA1. Heat induced activation elicits haustellum extension (middle panel, arrow). At the control temperature before (left panel) and after (right panel) activation the proboscis is retracted. (**B**) Heat induced silencing of GMR26A01 neurons using shibire[ts]. Flies at the permissive temperature show full PER upon

*Figure 6 continued on next page*

*Figure 6 continued*

200 mM sucrose stimulation (left and right panel). At the restrictive temperature, these flies fail to extend the haustellum (middle panel, arrow). (**C**) Expression pattern of GMR26A01 (mCD8-GFP, green) in the adult central brain (Brp, magenta). (**D**) Suppression of cholinergic expression using cha-Gal80. This intersectional strategy restricts expression to a single bilateral pair of MNs (arrow points to the axon). (**E**) Whole head preparation of GMR26A01, chaGal80 > mCD8-GFP demonstrates innervation of muscle 2 (asterisk). The boxed region is magnified in the right panels. Muscles are visualized by the F-actin marker phalloidin (blue). (**F**) Schematic drawing of the head muscles with innervated muscles highlighted in blue. Scale bars, 50 μm. (**G**) Quantification of the behavioral phenotypes in control and experimental animals. Numbers and significances are listed in *Supplementary file 1*. See also *Figure 5—figure supplement 1*, *Figure 6—figure supplement 1* and *Videos 6*, *7*, *8*, *9* and *10*.

The following figure supplement is available for figure 6:

**Figure supplement 1.** Unilateral proboscis MN activation induces asymmetric proboscis movement.

---

Analysis of GMR18B07, repoGal80>mCD8-GFP flies revealed one MN pair innervating muscle 9, and another MN pair innervating muscle 8 (*Figure 5F,G*). To validate our behavior results we repeated all behavior experiments in the presence of repo-Gal80. We observed identical results in these animals (*Figure 5A–C,H* and *Videos 3–5*). These results confirm the previously described role of MN9 and muscle 9 for rostrum lifting (*Gordon and Scott, 2009*) and identify MN8 and muscle 8 as potential regulators of labella spreading.

## MNs controlling haustellum extension

During sucrose-mediated activation of proboscis extension the lifting of the rostrum is followed by haustellum extension (folding down of the haustellum). In our functional screen using TrpA1 mediated activation we identified the line GMR26A01 as sufficient to induce a constant extension of the haustellum (*Figure 6A,G* and *Video 6*). Light-induced activation using Chrimson confirmed these results with the extension of the haustellum precisely correlating with the on and off-times of the red light stimulus (*Figure 6G* and *Video 7*). Consistent with these neurons controlling haustellum extension acute inhibition (GMR26A01>shibire^ts) prevented extension of the haustellum at the restrictive temperature in response to sucrose stimulation (*Figure 6B* middle panel,G and *Video 8*). However, these flies were still able to lift their rostrum and to extend and spread their labella (*Figure 6B*). The failure to extend the haustellum was fully reversible as flies completely extended their proboscis after reversal to the permissive temperature (*Figure 6B* and *Video 8*). Analysis of the expression pattern of GMR26A01-Gal4 revealed expression in 8–10 SEZ neurons (*Figure 6C*). While MNs in Drosophila are mainly glutamatergic the majority of excitatory neurons in the brain are cholinergic. To restrict expression to MNs we performed an intersectional genetic approach and co-expressed Gal80 selectively in all cholinergic neurons (cha-Gal80; *Kitamoto, 2002*) (GMR26A01, cha-Gal80>mCD8-GFP). These experiments restricted mCD8-GFP expression to a single pair of bilateral MNs with axons extending to the proboscis musculature (*Figure 6D,E*). The whole head preparation revealed that the MNs innervate muscle 2–1 (*Figure 6E,F*). Likely due to leakiness of the cha-Gal80 line, expression levels in MN 2–1 were strongly reduced (data not shown); however, in a small number of cases artificial activation using Chrimson (GMR26A01, chaGal80>Chrimson) was still sufficient to induce haustellum extension (*Figure 6G* and *Video 9*). Interestingly, in two flies we observed extension of the haustellum to either the left or the right side (*Figure 6—figure supplement 1A,B* and *Video 10*). Analysis of the

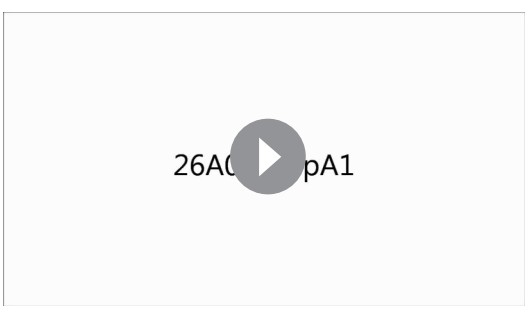

**Video 6.** Activation of GMR26A01 neurons using TrpA1 (related to *Figure 6*). This video shows side view sequences of a GMR26A01 > TrpA1 fly at 22°C, at 29°C, and at 22°C.

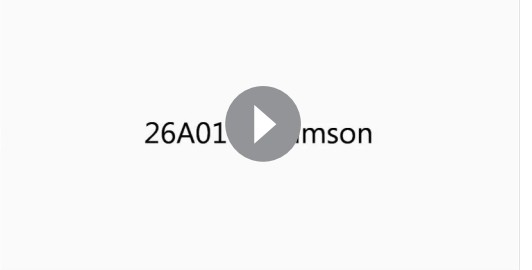

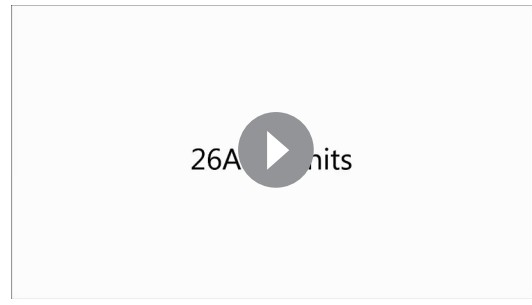

**Video 7.** Activation of GMR26A01 neurons using Chrimson (related to *Figure 6*). This video shows a side view sequence of a GMR26A01 > Chrimson fly at control, activation and control wavelength.

**Video 8.** Silencing of GMR26A01 neurons using shi[ts] (related to *Figure 6*). This video shows sucrose stimulations of a GMR26A01 > shi[ts] fly first at 22°C, at 29°C, and at 22°C, displayed at 0.5 x speed.

Chrimson expression pattern revealed a unilateral expression in the ipsilateral MN2 correlating with the direction of the haustellum extension (*Figure 6—figure supplement 1C*). Together these data indicate that MN2 controls the extension of the haustellum via activation of muscle 2–1.

## MNs controlling labella extension

Prior analysis of the proboscis extension sequence indicated that rostrum lifting and haustellum extension is followed by the spreading of the labella to enable food ingestion. Here, we identify extension of the labella as an additional step in the motor sequence that precedes labella spreading. Artificial activation of GMR81B12 expressing neurons (GMR81B12>TrpA1) resulted in a constant extension of the labella at the activation but not at the control temperature (*Figure 7A,G* and *Video 11*). Forward movement of the labella was particularly evident when using light-induced activation (GMR81B12>Chrimson; *Video 12*; *Figure 7G*). Acute silencing of GMR81B12 neurons (GMR81B12>shibire[ts]) during sucrose-mediated activation of the PER demonstrated that these neurons are not only sufficient but also required for labella extension (*Figure 7B,G* and *Video 13*). Analysis of the expression pattern revealed that GMR81B12-Gal4 is expressed within a single neuron in each brain-hemisphere (*Figure 7C*). This neuron innervates muscle 6 that is attached to the base of the labella (*Figure 7E,F*). The Gal4-expression within a single MN pair enabled us to determine the extent of dendritic versus axonal neurite arborization within the SEZ. To mark the dendritic compartment, we co-expressed the mCherry-tagged dendritic marker DenMark (*Nicolaï et al., 2010*) with

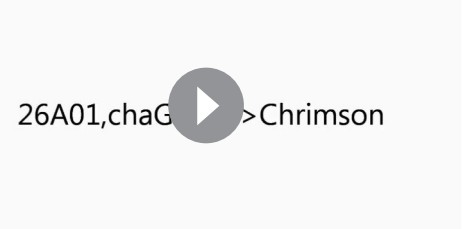

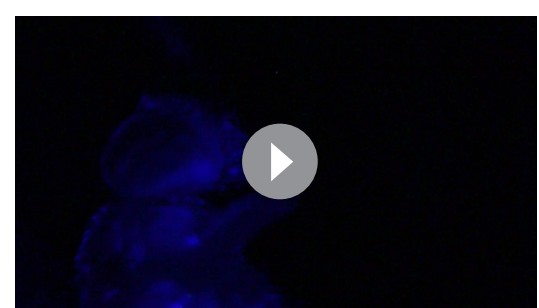

**Video 9.** Activation of GMR26A01, chaGal80 neurons using Chrimson (related to *Figure 6*). This video shows a side view sequence of a GMR26A01, chaGal80 > Chrimson fly at the control wavelength (475 nm), then at the activation wavelength (633 nm), and at the control wavelength.

**Video 10.** Activation of a unilateral GMR26A01, chaGal80 neuron using Chrimson (related to *Figure 6—figure supplement 1*). This video shows a side view and front view sequence of a GMR26A01, chaGal80 > Chrimson fly at control, activation and control wavelength.

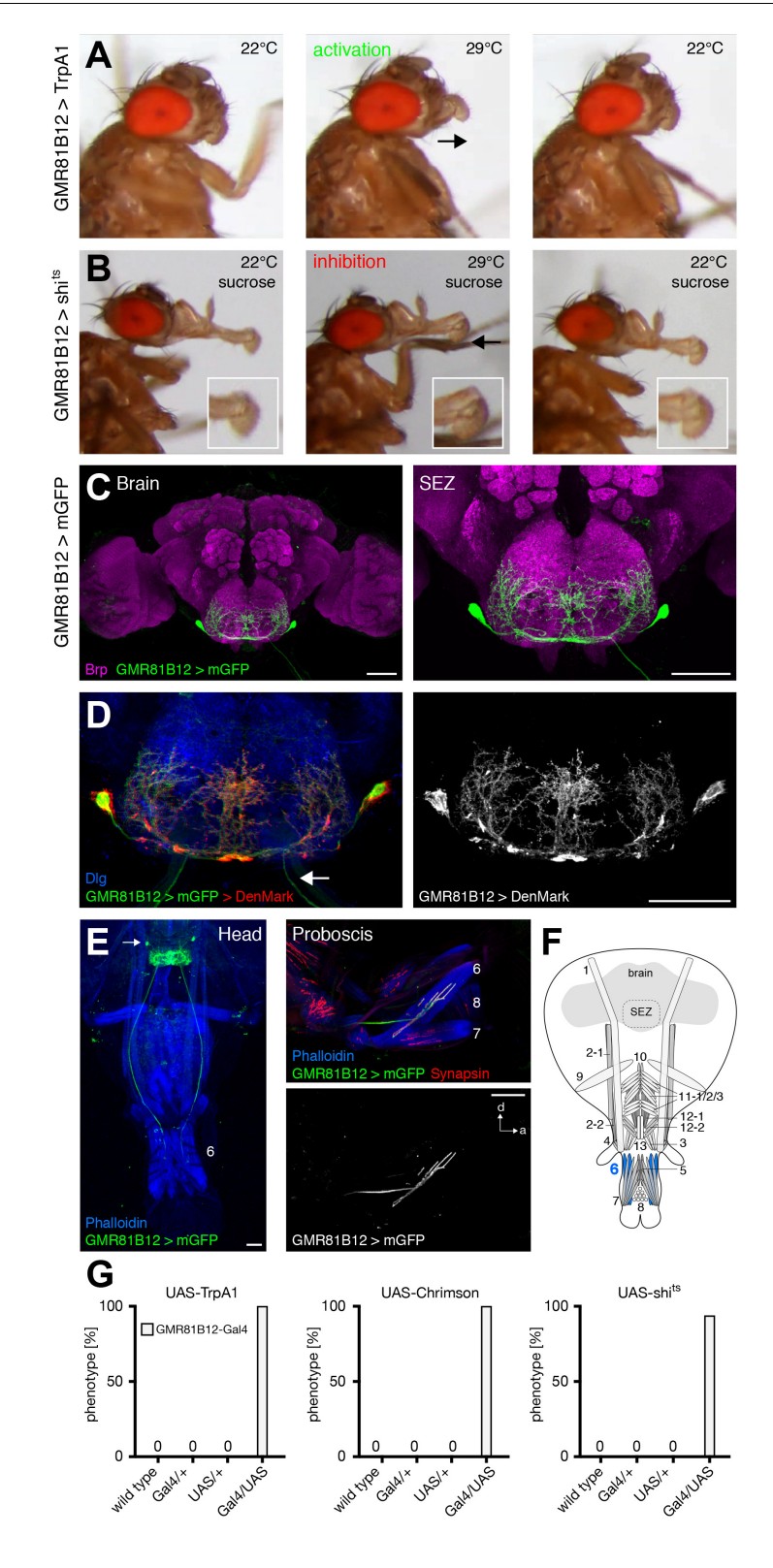

**Figure 7.** GMR81B12 neurons are sufficient and required for labella extension. (**A**) Artificial activation of GMR81B12 neurons using TrpA1. Heat induced activation elicits labella extension (middle panel, arrow). At the control temperature before (left panel) and after (right panel) activation the proboscis is retracted. (**B**) Heat induced silencing of GMR81B12 neurons using shibire[ts]. Flies at the permissive temperature show full PER upon 200 mM sucrose stimulation (left and right panel). At the restrictive temperature, these flies fail to extend the labella (middle panel, arrow). Insets show magnifications of

*Figure 7 continued on next page*

*Figure 7 continued*

the labella. (**C**) Expression of GMR81B12 (mCD8-GFP, green) in the adult central brain reveals a single MN in each brain hemisphere. (**D**) Analysis of dendritic versus axonal arborization. The mCherry-tagged dendritic marker DenMark (red, left panel; white, right panel) co-localizes with the general membrane marker mCD8-GFP (green) in the SEZ but not in the MN-axons projecting out of the brain (arrow). The neuropil is marked by Dlg (blue). (**E**) Whole head and proboscis preparation of GMR81B12 > mCD8-GFP flies reveals that the MNs (green, arrow indicates MN cell body) innervate muscle 6. Muscles are visualized by the F-actin marker phalloidin (blue) and NMJs are labeled using the presynaptic vesicle marker Synapsin (red). (**F**) Schematic drawing of the head muscles with innervated muscles highlighted in blue. Scale bars, 50 μm. (**G**) Quantification of the behavioral phenotypes in control and experimental animals. Numbers and significances are listed in *Supplementary file 1*. See also *Figure 5—figure supplement 1* and *Videos 11* , *12* and *13*.

the general membrane marker mCD8-GFP. Within the SEZ DenMark completely co-localized with mCD8-GFP indicating that the entire SEZ arborization is of dendritic nature (*Figure 7D*). The MN axons projecting through the labial nerve lacked any DenMark expression demonstrating the specificity of the marker. These results demonstrate that MN6 controls extension of the labella via activation of muscle 6.

## MNs controlling labella spreading

The analysis of GMR18B07-Gal4 revealed that artificial activation of muscle 8 via MN8 is sufficient to induce spreading of the labella (*Figure 5*). We identified two additional lines, GMR58H01 and VT020958, that induced spreading of the labella upon artificial activation with either TrpA1 or Chrimson (*Figure 8A,C,G* and *Videos 14* and *15*; *Figure 8—figure supplement 1A,F*). While both lines are expressed in multiple MNs (*Figure 8D–F*, *Figure 8—figure supplement 1C–E*) the only common MN between the three lines is MN8 suggesting that activation of muscle 8 controls labella spreading. However, silencing of these neurons was not sufficient to prevent labella spreading upon sucrose stimulation (*Supplementary file 1*). This failure to impair labella spreading is likely due to insufficient inhibition of the MN. Together, these results suggest that MN8 controls labella spreading but we cannot formally rule out the contribution of additional MNs. Consistent with the expression of line VT020958 in labella muscles 6 and 8 artificial activation induced not only labella spreading but also labella extension verifying the role of MN6 (*Figure 8—figure supplement 1B–E*). Artificial activation did not reveal a role for MN7 that is also targeted by line GMR58H01 (*Figure 8*, *Figure 8—figure supplement 1*).

## MNs controlling proboscis retraction

Artificial activation of neurons of line GMR58H01 did not only resulted in labella spreading but at the same time caused a retraction of the proboscis into the head capsule (*Figure 8B,C*, *Videos 14* and *16*). To directly test a potential contribution of GMR58H01 MNs to proboscis retraction we combined Chrimson-mediated activation with sucrose induced proboscis extension. Under control condi-

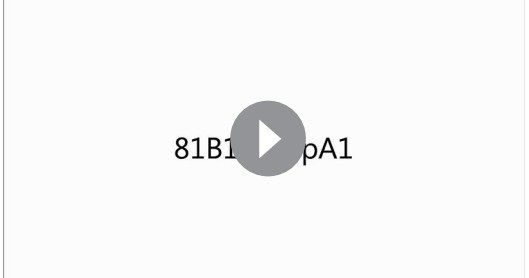

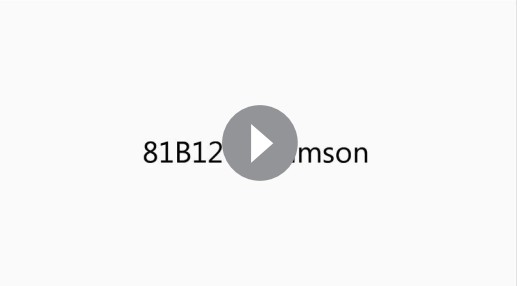

**Video 11.** Activation of GMR81B12 neurons using TrpA1 (related to *Figure 7*). This video shows side view sequences of a GMR81B12 > TrpA1 fly at 22°C, at 29°C, and at 22°C.

**Video 12.** Activation of GMR81B12 neurons using Chrimson (related to *Figure 7*). This video shows a side view sequence of a GMR81B12 > Chrimson fly at control, activation and control wavelength.

tions (blue light) sucrose stimulation of fly legs induced complete proboscis extension (*Figure 8C*, *Video 16*). In contrast, under activation conditions (red light) these flies failed to extend their proboscis in response to sucrose stimulation (*Figure 8C*, *Video 16*; note also MN8 dependent labella spreading). Analysis of the expression pattern of GMR58H01 revealed selective expression in 4 MNs, MN1, MN4, MN7 and MN8 (*Figure 8D–F*). Based on morphology and cross-comparison to the other MN lines we can exclude MN4, 7 and 8 indicating that MN1 likely mediates active retraction of the proboscis into the head capsule. Indeed, such a function has been previously suggested for MN1 in blowflies (*van der Starre and Ruigrok, 1980*). Silencing of GMR58H01 neurons including MN1 did not significantly impair retraction of the proboscis after

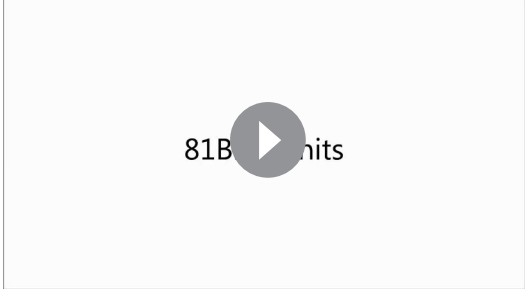

**Video 13.** Silencing of GMR81B12 neurons using shi[ts] (related to *Figure 8*). This video shows sucrose stimulations of a GMR81B12 > shi[ts] fly at 22°C, at 29°C, and at 22°C, displayed at 0.5 x or 0.125 x speed.

sucrose stimulation. In contrast to muscle 8 that is innervated by a single MN our MARCM data revealed that muscle 1 is innervated by multiple MNs (*Figure 2E*) and inhibition of a single MN is likely not sufficient to prevent muscle contraction. Alternatively, additional muscles may participate in proboscis retraction.

## Step-wise control of proboscis extension and retraction

The identification and genetic control of the MNs controlling five major steps of proboscis extension and retraction, lifting of the rostrum (MN9), extension of the haustellum (MN2), extension of the labella (MN6), spreading of the labella (MN8) and proboscis retraction (MN1) enabled us to next address the neuronal circuit architecture controlling the motor pattern. In general, two alternative principles could generate the observed fixed sequence of events. In a first model, the PER is based on a chain reflex sequence in which the initiation of each step depends on the successful execution of the preceding step of the motor sequence. Alternatively, all steps are independently initiated and coordinated at the level of pre-motor interneurons. To address these alternative hypotheses, we first analyzed the proboscis extension sequence of flies in which single MNs were silenced while applying positive taste stimuli. In a second step, we performed corresponding experiments in which we artificially activated MNs while applying positive taste stimuli. Single image analysis of the recorded sequences of our silencing experiments demonstrated that subsequent steps of the motor sequence could be efficiently executed despite the failure to perform a central step of the serial sequence (*Figures 5*, *6*, *7* and *8*, *Figure 5—figure supplement 1* and *Figure 8—figure supplement 1*). For example, despite complete inhibition of rostrum lifting (MN9 silencing) flies were still able to extend the haustellum and labella (*Figure 5B* and *Video 5*). Similarly, blocking haustellum extension did not prevent extension or spreading of labella (*Figure 6B* and *Video 8*). The only exception from this rule was observed in flies where we blocked labella extension. Here, sucrose stimulation of legs was no longer sufficient to induce labella spreading (*Video 13* and *Figure 5—figure supplement 1*). However, direct sucrose stimulation of gustatory sensory sensilla present on the labella reliably elicited labella spreading in these flies. Thus, despite inappropriate positioning of individual proboscis elements the consecutive steps of the motor sequence were efficiently executed. In contrast, analysis of the temporal profiles of individual sequence steps revealed significant alterations in these flies. In control flies sucrose stimulation induces a rapid progression through the PER sequence (*Figure 1G*, *Figure 5—figure supplement 1A*). Inhibition of rostrum lifting significantly prolonged the time from stimulus to haustellum extension but accelerated progression from labella extension to labella spreading (*Figure 5—figure supplement 1B*). Inhibition of haustellum extension significantly reduced the time from rostrum lifting to labella extension and from labella extension to labella spreading (*Figure 5—figure supplement 1C*). Inhibition of labella extension not only perturbed progression to labella spreading but also increased the time duration from stimulus to rostrum lifting. These data indicate that serial execution of the individual movements is necessary to achieve the temporal precision observed in wild type flies. A potential explanation for the majority of the

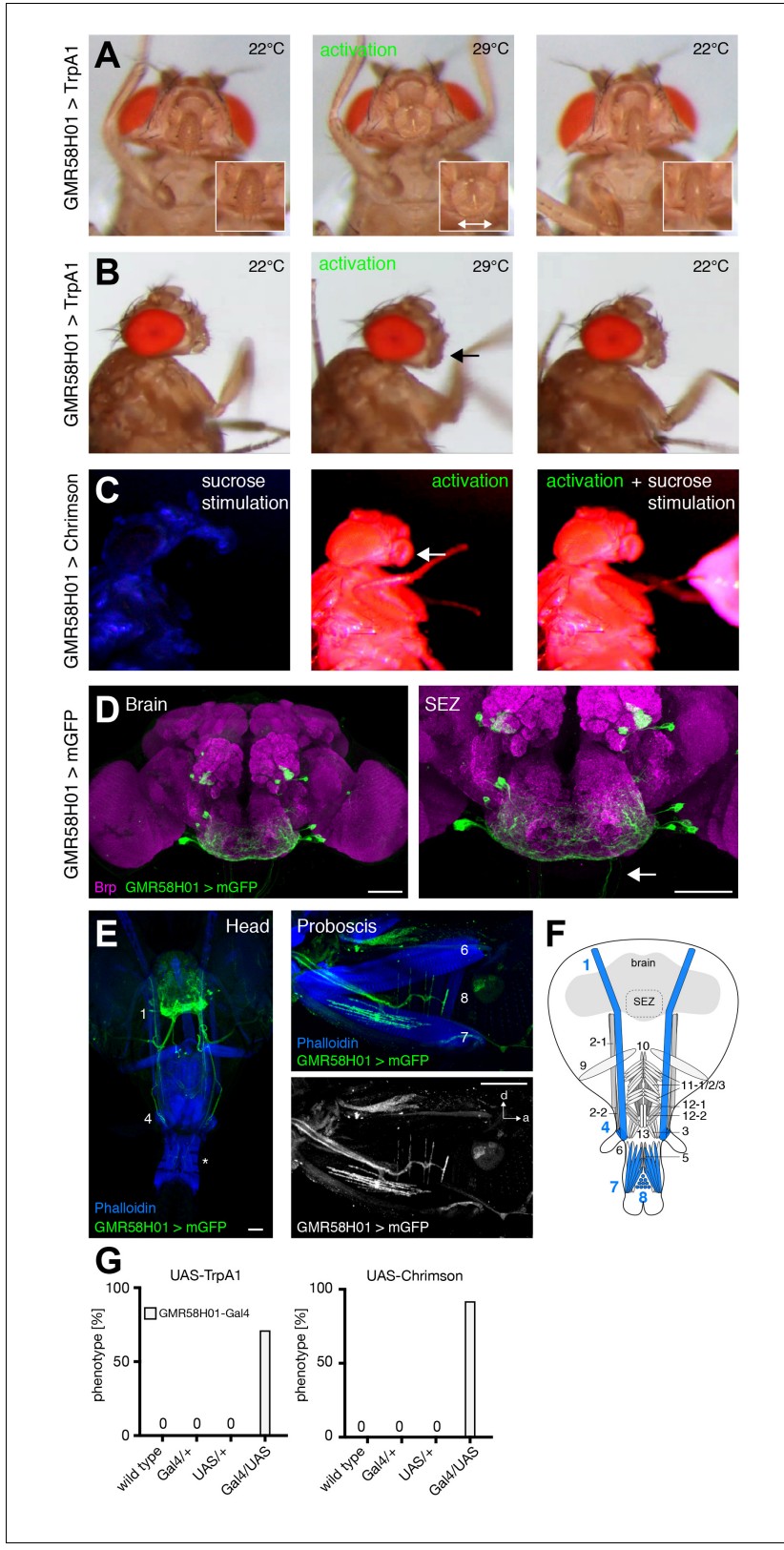

**Figure 8.** GMR58H01 neurons elicit labella spreading and proboscis retraction. (**A**,**B**) Artificial activation of GMR58H01 neurons using TrpA1. Heat induced activation elicits labella spreading (middle panel (**A**)) and leads to the retraction of the proboscis (middle panel (**B**), arrow). At the control temperature before (left panels) and after (right panels) activation the proboscis is retracted. Insets in (**A**) show magnifications of the labella and the double

*Figure 8 continued on next page*

*Figure 8 continued*

arrow indicates the spreading of the labella. (**C**) Artificially activation of GMR58H01 neurons using Chrimson while evoking sucrose induced proboscis extension. At blue light (control), flies show full PER upon 200 mM sucrose stimulation (left panel). Red light activation results in labella spreading and proboscis extension (middle panel). Red light activation during 200 mM sucrose stimulation prevents proboscis extension (right panel). (**D**) Expression pattern of GMR58H01 in the adult central brain. Arrow points to the axons that are leaving the brain. (**E**) Whole head preparation of GMR58H01>mCD8-GFP flies (left panel) reveals that the identified MNs (green) innervate muscle 1, 4, and haustellum muscles (asterisk). The side view of the proboscis (right panels) shows innervation of muscles 7 and 8 in the haustellum. Muscles are visualized by the F-actin marker phalloidin (blue). (**F**) Schematic drawing of the head muscles with innervated muscles highlighted in blue. Scale bars, 50 μm. (**G**) Quantification of the behavioral phenotypes in control and experimental animals. Numbers and significances are listed in ***Supplementary file 1***. See also ***Figure 8—figure supplement 1***, ***Videos 14***, ***15*** and ***16***.

The following figure supplement is available for figure 8:

**Figure supplement 1.** VT020958 neurons elicit labella spreading.

observed alterations may be anatomical constrains of the movement, however we cannot rule out that sensory feedback mechanisms contribute to the robustness of the motion sequence.

We next analyzed whether artificial activation of individual MNs would impair the normal extension response elicited by positive (sweet) stimulation of the gustatory sensory neurons on the forelegs. As a readout, we measured the maximum proboscis extension distance in response to sucrose stimulation in control flies and in flies with artificially activated MNs (***Figure 9***, see Materials and methods). We first applied this method to line GMR58H01 (MN1, 4, 7, 8) to quantify the consequences of activation of the retractor MN1. Artificial activation (via TrpA1 or Chrimson) of MN1 almost completely prevented proboscis extension in response to the sweet sensory stimulus despite normal displacement at the permissive temperature and under blue light exposure (***Figure 9A***). Indeed, just activation of line GMR58H01 induced a retraction of the proboscis into the head capsule resulting in a small but significant negative extension value (***Figure 9A***). In contrast, activation of MN6 (line GMR81B12, labella extension) did not significantly alter sucrose evoked proboscis extension distance (***Figure 9B,F***). However, artificial activation of both line VT020958 (MN2, 6, 7, 8; labella extension and spreading) and of line GMR18B07 (MN9, 8; activation of rostrum lifting and labella spreading) significantly reduced the maximum proboscis extension distance (***Figure 9C, D,F***). These experiments demonstrate that full extension of the proboscis is not achieved by additive complete contractions of participating muscle groups but requires a precise temporal coordination of activation intensities.

## MN based control of the proboscis extension response

Based on these results we propose that 5 MNs control the major steps of proboscis extension and retraction (***Figure 10***). Upon a positive gustatory stimulus flies first lift the rostrum (MN9), extend the haustellum (MN2), extend the labella (MN6), spread the labella for food ingestion (MN8) and finally retract the proboscis (MN1) (***Figure 10***). Analysis of the dendritic arborizations of these MNs revealed a stereotypic organization within the SEZ with all MN dendrites sharing a common space that mainly occupies the anteroventral regions of the SEZ with two spared 'ball like-structures' on both sides of the midline (***Figure 10B–F***, right panels; ***Figure 10— figure supplement 1A–C***). It has been previously reported that MN9 is not directly connected to gustatory sensory neurons (***Gordon and Scott,***

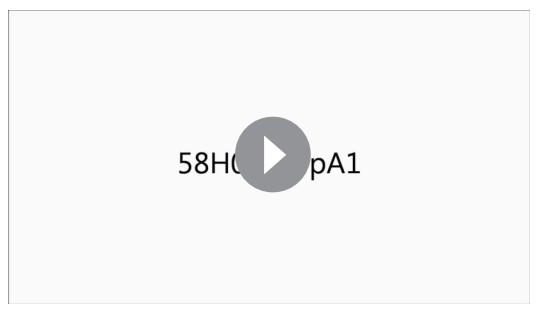

**Video 14.** Activation of GMR58H01 neurons using TrpA1 (related to ***Figure 8***). This video shows front view sequences of a GMR58H01 > TrpA1 fly at 22°C, at 29°C, and at 22°C.

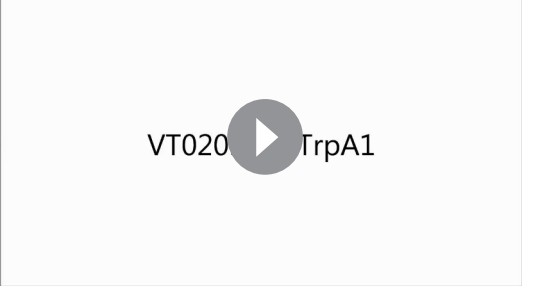

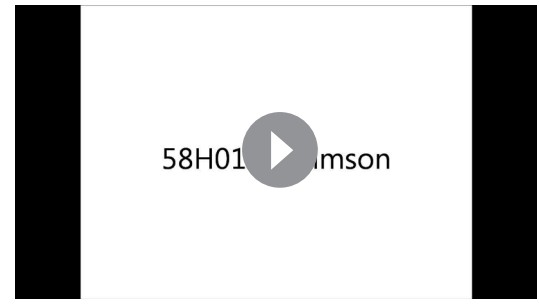

**Video 15.** Activation of VT020958 Neurons using TrpA1 (related to *Figure 8—figure supplement 1*). This video shows sequences of a VT020958 > TrpA1 at control and activation temperatures. Order of video sequences: Front view at 22°C, front view at the transition to 29°C, front view at 22°C, side view at 22°C, side view at 29°C, and side view at 22°C.

**Video 16.** Activation of GMR58H01 neurons using Chrimson (related to *Figure 8*). This video shows sucrose stimulations of a GMR58H01 > Chrimson fly at the control wavelength (475) and then at the activation wavelength (633 nm).

*2009*) that innervate posterior-dorsal regions of the SEZ (*Figure 10—figure supplement 1D*). To determine whether this observation holds true for all MNs we investigated potential connectivity of our MN lines to Gr5a-expressing sweet gustatory sensory neurons using the same GRASP approach (*Feinberg et al., 2008*). In these experiments, we did not observe significant GRASP signals (*Figure 10—figure supplement 2*). As a positive control, we observed strong GRASP signals between Gr5a-positive sensory neurons and inhibitory interneurons (*gad1*-Gal4; *Sakai et al., 2009*). Thus, control of the proboscis motor sequence is likely mediated via a dedicated set of interneurons downstream of gustatory sensory neurons.

Finally, we performed MN co-labelling experiments to investigate the spatial relationship of MN dendrites within the SEZ. We utilized a LexA-version of our MN6-Gal4 line (GMR81B12-LexA = MN6-LexA) that shows co-labelling of the soma and dendrites with the MN6-Gal4 line (*Figure 11A*). Simultaneous labelling of MN2 (GMR26A01-Gal4, cha-Gal80) and MN6 revealed largely overlapping dendritic arborization patterns with more extensive arborization of MN6 at the midline region (*Figure 11B*). The high regional overlap of dendrites of distinct identity was particularly evident when the dendritic arborization of MN6 was analyzed together with MNs 1,4,7 and 8 (GMR58H01-Gal4) (*Figure 11C*). In single sections a close but non-overlapping association of MN dendrites can be observed (*Figure 11B,C* lower panels).

## Discussion

In this study, we provide a comprehensive developmental, neuroanatomical and functional characterization of the MNs controlling proboscis extension and retraction. We demonstrate that four MN types control the four major steps of proboscis extension while one MN likely contributes to the active retraction of the proboscis. These temporally ordered steps are independently controlled in a one-to-one manner with the majority of MNs both sufficient and required for the execution of one individual step of the forward reaching behavior. Our data demonstrate that MN-based feed-forward activation does not contribute to the precise temporal control of proboscis motion. Coupling of individual motor steps likely occurs at the level of premotor interneurons that provide the basis for selective execution of different motor subprograms of proboscis motion required during innate behaviors including courtship and gustatory behavior.

### Organization and origin of proboscis motoneurons

Our MARCM-based single cell clonal analysis shows that the different types of proboscis MNs can be divided into two major groups that differ in terms of cell body position, dendritic arborization, axonal projection and muscle innervation pattern. The first group comprises seven MN types innervating muscles 1, 2, 3, 4, 6, 7, and 8. These MNs are bilaterally symmetric and their entire dendritic

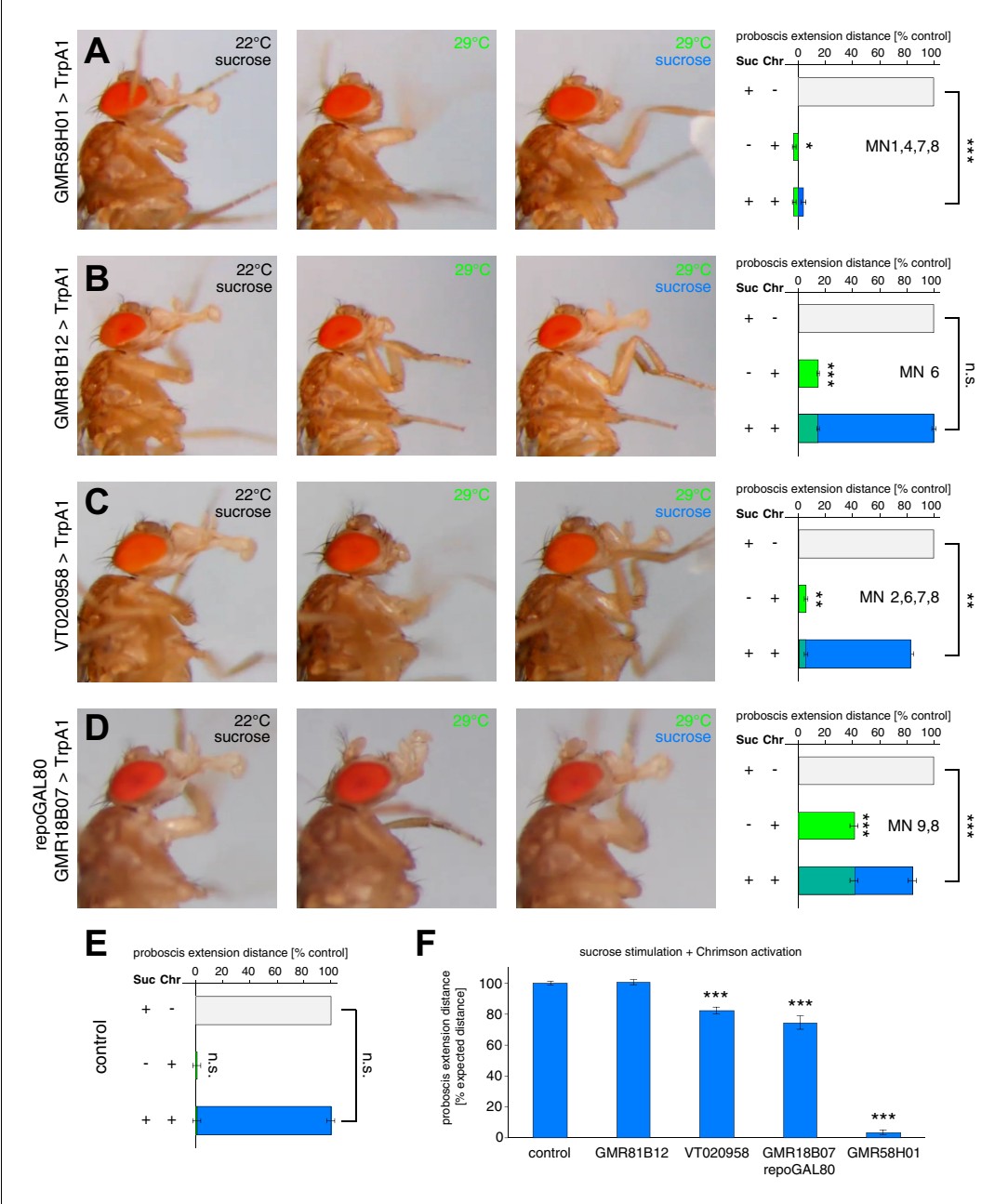

**Figure 9.** The PER motor program relies on the precise coordination of motoneuron activity. (A–D) MN-Gal4>TrpA1 and MN-Gal4>Chrimson animals were used to display (TrpA1) and quantify (Chrimson) the effects of constant MN activation during sucrose evoked proboscis extension. Animals were stimulated with 200 mM sucrose and snapshots of the maximum proboscis displacement at control (panel 1) and activation temperature (panel 3) are shown. Proboscis displacement in response to heat induced activation is shown in panel 2. Graphs: Maximum proboscis displacement was measured at blue light (control) upon 200 mM sucrose stimulation (grey bar), at red light (activation) without sucrose stimulation (green bar), and at red light upon 200 mM sucrose stimulation (green + blue bar). The zero point is defined by the position of the proboscis at blue light without sucrose stimulation and 100% proboscis extension represents the maximum proboscis displacement at blue light upon 200 mM sucrose stimulation. Data are presented as mean ± SEM. (E) Same quantification as in (A–D) for control (w[1118]>Chrimson) animals. (F) Quantification of percentage of expected proboscis distance. The zero point is defined by the proboscis displacement at red light without sucrose stimulation while 100% proboscis extension represents maximum proboscis displacement at blue light upon 200 mM sucrose stimulation. Data are presented as mean ± SEM. Statistical analysis: see Materials and methods.

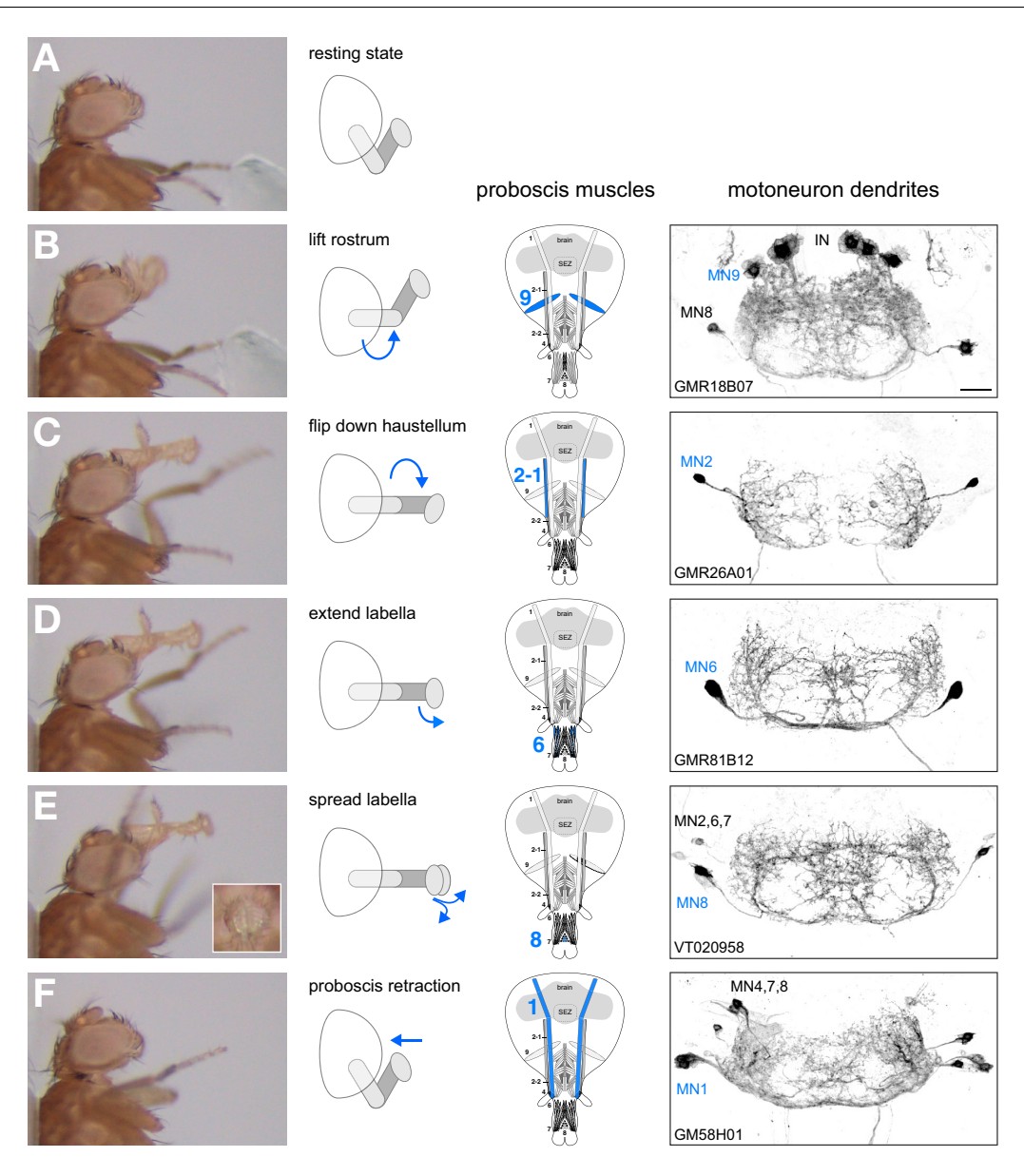

**Figure 10.** Motoneurons and muscles controlling the major steps of the PER motor program. The extension of the proboscis in response to an attractive stimulus (200 mM sucrose, **A**–**E**) follows a stereotypic pattern that can be subdivided into a sequence of events. Snapshots of the motion pattern of a control fly (w[1118]) executing a continuous proboscis extension (left panels) and in the schematic drawings illustrating the direction of the movements (blue arrows) are shown. Muscles and MNs controlling the individual steps are indicated in the schematics (blue numbers) and in brains (right panels, inverted fluorescence images from *Figures 5–8*). MNs display a striking overlap in dendritic organization. Scale bar, 20 µm.

The following figure supplements are available for figure 10:

**Figure supplement 1.** Spatial organization of MN dendrites in the SEZ.

**Figure supplement 2.** Gr5a sensory neurons do not form synaptic connections with proboscis MNs.

arborization is restricted to the anteroventral SEZ. Within each hemi-ganglion the MN cell bodies are clustered together, axons project through the labial nerve and they innervate ipsilateral muscle groups with respect to their cell body position. The second group comprises four MN types innervating muscle groups 5, 10, 11 and 12 via the pharyngeal nerve. Strikingly, in contrast to the first group

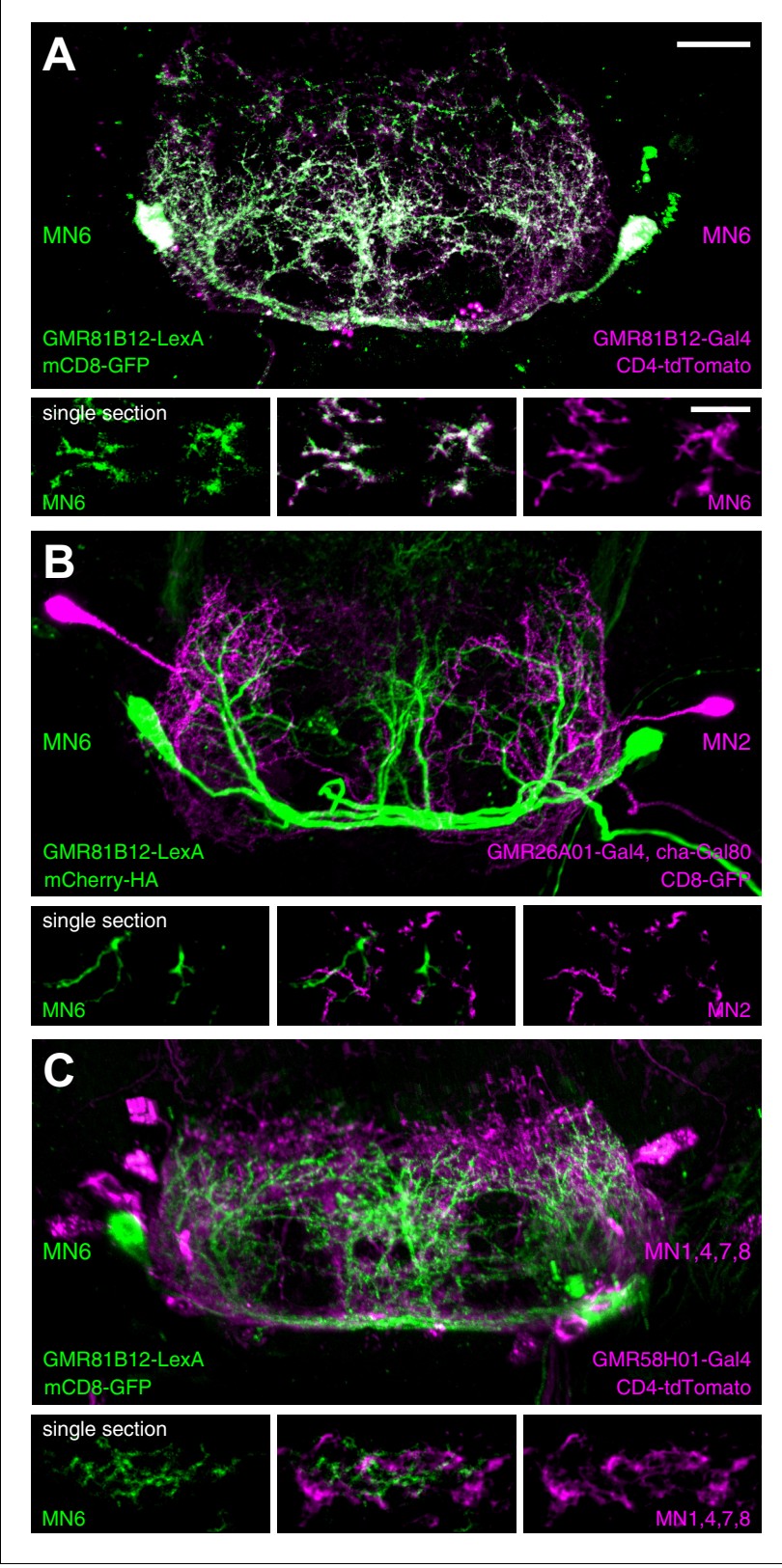

**Figure 11.** Analysis of dendritic arborizations of proboscis MNs. (**A–C**) Coexpression of mCD8-GFP and tdTomato/mCherry in MN-expressing lines using two binary expression systems (Gal4 and LexA). Maximum intensity projections (upper panel) and single z-stack sections (lower panel) are displayed. Spatial relationship

*Figure 11 continued on next page*

*Figure 11 continued*
between GMR81B12-LexA targeted MN6 (green) and one of the following MN-lines are shown: GMR81B12-Gal4 targeted MN6 (**A**, magenta), GMR26A01-Gal4, cha-Gal80 targeted MN2 (**B**, magenta), and GMR81B12-Gal4 targeted MNs1, 4, 7, 8 (**C**, magenta). Single sections in **B**, **C** show close, but non-overlapping association of MN dendrites of different MN populations. Scale bar, 20 µm (overview), 10 µm (single sections).

the axons of these MNs bifurcate and simultaneously innervate homologous muscles on both the ipsi- and contralateral side. The only exception to these rules is MN9 that based on its ipsilateral innervation of muscle 9 belongs to group 1, however its cell body clusters with group 2 MNs and it projects via the pharyngeal nerve to the proboscis.

The neuroanatomical features of the two MN groups directly reflect their unique and different functional roles. Group 2 MNs control muscle groups (5, 10, 11, and 12) that elicit the rhythmic and bilaterally symmetric activity of the pharyngeal pump required for food ingestion (*Dethier, 1976*; *Miller, 1950*). Our data now demonstrate that the axons of these MNs bifurcate and provide equal input to target muscles on the ipsi- and contralateral site. As, in addition, the dendritic arborizations are equally distributed within both hemispheres any stimulatory input (frontal, left or right) will be translated into a symmetric activation of pharyngeal pump muscles to ensure appropriate food uptake.

In contrast, group 1 MNs control muscle groups that mediate the extension, retraction and positioning of the proboscis (muscle groups 1, 2, 3, 4, 6, 7, 8 and 9). Our analysis demonstrates that all these MNs innervate ipsilateral located muscles but differ in their dendritic arborization patterns within the SEZ. Some MNs have predominantly ipsilateral while others have predominantly contralateral dendritic arborizations. All group 1 MN dendrites are restricted to the anteroventral SEZ region with dendrites of different MNs often present in close proximity to each other (*Figure 11*). This highly elaborate dendritic organization likely enables a direct translation of side-specific stimulation into directed movement. Indeed, similar to prior observations in the blowfly (*Yetman and Pollack, 1987*) sucrose stimulations of one leg induce proboscis extension towards the stimulus direction (*Video 17*). Furthermore, selective activation of an individual MN2 induced the extension of the haustellum towards the activation side (*Figure 6—figure supplement 1*). Together, our analysis revealed a remarkable level of hard-wired organization to accommodate the specific tasks of direction-selective and direction-independent MNs.

The fly proboscis is an appendage of the head composed of highly reduced and bilaterally fused mouthparts that represents a serial homolog of other segmental appendages such as the thoracic legs. It is thus interesting to consider possible homologies between proboscis and thoracic leg MNs. The MNs innervating the prothoracic leg have been well characterized and comprise 53 MNs that derive from 11 independent neuroblasts, with two lineages giving rise to 35 of the 53 MNs (*Baek and Mann, 2009*; *Brierley et al., 2012*). Most of these MNs are generated postembryonically during larval development and match the development of the leg (*Estella and Mann, 2008*; *Estella et al., 2008*; *McKay et al., 2009*; *Morata, 2001*; *Soler et al., 2004*). In contrast, a hemi-proboscis is only innervated by approximately 20 MNs, and our MARCM labeling demonstrated that all MNs are born during embryogenesis (0–12 hr AEL). In contrast to the leg MNs, individual labeled clones never included more than one type of proboscis MN suggesting that each of the thirteen different MN types are generated by different neuroblasts. The fact that proboscis MNs are generated during embryogenesis indicates that these MNs also have potential roles during larval stages. Indeed, it has been reported that MNs innervating the adult muscle

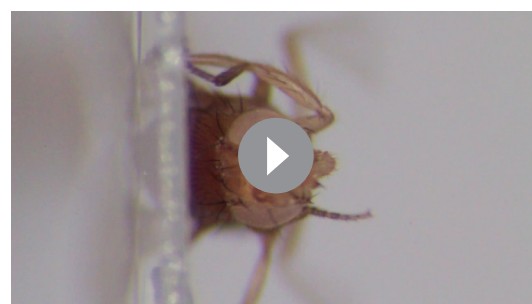

**Video 17.** Proboscis extension response after unilateral leg stimulation. This video shows a top view sequence of a unilateral sucrose stimulation to the right front leg first in real time followed by slow motion (0.32 x speed).

11 are also required for feeding in larvae (*Tissot et al., 1998*). Thus, proboscis MNs may provide analogous functions during larval food ingestion despite different functional requirements and body organization. It will be of great interest to determine the precise use of these MNs during larval development and to address the potential developmental mechanism underlying the morphological and functional reorganizations necessary to accommodate the different functional requirements.

## Motor control of the serial proboscis extension response

The detailed anatomical analysis of proboscis muscles and MNs together with the genetic manipulation of individual MNs enabled us to demonstrate that five MNs are sufficient to control the major steps of proboscis extension behavior. We show that gustatory stimulation elicits five consecutive, partially overlapping movements: rostrum lifting, haustellum extension, labella extension, labella spreading and proboscis retraction. Each of the steps is controlled by one bilateral pair of muscles that are innervated by one or multiple pairs of MNs. In all cases, artificial MN activation was sufficient to elicit a single step of the serial proboscis motion. In contrast, inhibition of MN activity did not always prevent execution of the specific movement. Three reasons can account for this observation: First, some muscles are innervated by multiple MNs and inhibition of a single MN is not sufficient to prevent muscle contraction as observed for muscle1. Second, the expression levels of the inhibitory construct may not always be sufficient to shut down MN activity. And third, we cannot exclude the possibility that additional muscles contribute to individual steps of the PER that would act at least partially redundant. Despite these limitations regarding the requirements of individual muscle groups our combined data clearly demonstrate that all steps of the motor sequence are individually controlled. Importantly not only proboscis extension but also proboscis retraction is potentially controlled by active mechanisms. Active termination of the PER likely contributes both to the repetitive PER behavior observed in vivo (*Itskov et al., 2014*, see below) and to aversive responses to bitter substances (active retraction of the proboscis, data not shown).

In addition, we provide evidence that initiation of individual movements does not depend on the execution of earlier steps of the motor sequence. However, a serial execution of the individual movements is necessary to achieve temporal precision of the PER sequence as inihibition of individual MNs led to perturbations of the stereotypic temporal profile of the PER motion. Three reasons may account for these alterations: First, the central control of serial activation of individual MNs may be perturbed by the artificial silencing of one of the target MNs. Second, the failure to move parts of the proboscis may affect sequence execution due to anatomical constraints. Third, altered sequence execution may affect sensory feedback systems as observed in the leg motor system of Drosophila (*Mendes et al., 2013*).

Using activation experiments we demonstrated that the execution of one movement does not automatically trigger the initiation of the subsequent movement. In contrast to a reflex chain as observed for crayfish escape behavior (*Reichert et al., 1981*) the movement of the proboscis elements is likely controlled in a one-to-one manner by individual MNs. Thus, our data indicates that the generation of the temporal proboscis motion sequence is programmed upstream of the MNs in the central brain. This central coordination of MN activity is consistent with the observation that different stereotypic proboscis movements are part of at least two additional innate behavior programs. During male courtship behavior the proboscis displays an upward motion that includes rostrum lifting, labella extension and labella spreading (courtship licking) (*Hall, 1994*; *Nichols et al., 2012*) while the proboscis is placed outwards of the head capsule during proboscis cleaning (*Hampel et al., 2015*; *Seeds et al., 2014*). It is thus likely, that these three innate proboscis motions, feeding, licking and grooming, are independently controlled by central circuits inducing context-specific motor unit recruitment profiles. This is supported by the observation that activation of an individual command interneuron is sufficient to induce the entire proboscis feeding motion (*Flood et al., 2013*). As this command neuron is not directly connected to MNs (*Flood et al., 2013*) the selective and sequential activation of the individual MNs requires at least an additional layer of interneurons. For peristaltic larval locomotion it has recently been demonstrated that the sequential and partially overlapping activation of intrasegmental MNs is controlled by both excitatory and inhibitory interneurons (*Zwart et al., 2016*). The MNs controlling distinct muscle groups are innervated by largely non-overlapping excitatory interneurons similar to observations in vertebrates (*Bikoff et al., 2016*; *Goetz et al., 2015*; *Tripodi et al., 2011*). Interestingly, however, the phasic motoneuron activation delay is mainly generated by selective inhibitory MN innervation

(*Zwart et al., 2016*). While we currently lack any information regarding the upstream interneurons controlling proboscis motion our data is consistent with either selective inhibition or excitation generating unique proboscis extension motions. For example, in contrast to the feeding motion the proboscis extension sequence during courtship licking lacks the haustellum extension step and consists only of rostrum lifting, labella extension and labella spreading. Finally, analysis of natural feeding behavior demonstrated that flies display rhythmic patterns of proboscis extension and retraction when feeding on gelatinous food but not when drinking liquids. Thus, depending on the food quality a CPG contributes to the control of repetitive proboscis extension (*Itskov et al., 2014*). The genetic control and simplicity of the underlying motor system will greatly facilitate the identification and characterization of cellular and circuit principles controlling this reaching-like motor sequence.

## Materials and methods

### Fly stocks

Fly stocks were maintained on standard fly food at 25°C. Crosses for immunohistochemistry were kept at 25°C, while crosses for neuronal activation and silencing experiments were kept at 22°C. Enhancer-Gal4 and -LexA lines were obtained from the Bloomington Drosophila Stock Center (*Jenett et al., 2012*) and the Vienna Drosophila RNAi Center (*Kvon et al., 2014*). The following fly strains were used in this study: $w^{1118}$, GMR18B07-*Gal4* (RRID:BDSC_47476), GMR26A01-*Gal4* (RRID: BDSC_49148), GMR81B12-*Gal4* (RRID:BDSC_40107), GMR58H01-*Gal4* (RRID:BDSC_39197), VT020958-*Gal4* (RRID:FlyBase_FBst0485173), GMR81B12-*LexA* (RRID:BDSC_54389) OK371-*Gal4* (RRID:BDSC_26160), *FRT19A/FM7c*, *FRT19A,hsFLP*,Tubulin-*Gal80; OK371-Gal4*,UAS-*mCD8-GFP/ CyO*, Gr5a-*LexA*;UAS-*tdTomato::LexAop2-CD4-spGFP11;UAS-CD4-spGFP1–10* (*Feinberg et al., 2008*; *Gordon and Scott, 2009*), Gad1-*Gal4* (RRID:BDSC_51630; *Sakai et al., 2009*), MHC-GFP (RRID:BDSC_38462; *Chen and Olson, 2001*), 5xUAS-*mCD8-GFP* (RRID:BDSC_32192), 10xUAS-*mCD8-GFP* (RRID:BDSC_32186), UAS-*CD4-tdTomato* (RRID:BDSC_35841), 13xLexAop2-*mCD8-GFP* (RRID:BDSC_32203), UAS-*DenMark* (RRID:BDSC_33061; *Nicolaï et al., 2010*), UAS-*TrpA1* (RRID: BDSC_26263; *Hamada et al., 2008*), UAS-*Chrimson* (RRID:BDSC_55135; *Klapoetke et al., 2014*), UAS-*shibire*$^{ts}$ (*Kitamoto, 2001*), cha-Gal80 (*Kitamoto, 2002*), repo-Gal80 (*Awasaki et al., 2008*).

### Backfilling of motoneuron nerves

To label all the MNs innervating the proboscis, flies with the genotype *OK371-Gal4*,UAS-*mCD8GFP* were used. The proboscis was cut from the tip of the head and a crystal of rhodamine-labelled dextran dye was placed on cut nerves. The dye was left to diffuse for 4 hr at 4°C. The brain was then dissected, fixed, washed and mounted as described below.

### MARCM analysis

To label individual MNs, single cell MARCM clones were induced during embryonic or post embryonic neurogenesis. For these experiments, females of the genotype *FRT19A/FM7c* were crossed with males of the genotype *FRT19A,hsFLP*,Tubulin-*Gal80; OK371-Gal4*,UAS-*mCD8-GFP/CyO*. For clone induction during embryogenesis, embryos were collected for 4 hr at 25°C and heat shocks were applied for 1 hr at 37°C at different time points. Similarly, for post embryonic clone induction larvae were collected at different time intervals from 24 hr after larval hatching (ALH) to 96 hr ALH and heat shocks were applied after different time points.

### Immunohistochemistry of MARCM samples

Dissections of adult brains with the proboscis were carried out in 1x phosphate-buffered saline (PBS) and fixed in 4% freshly prepared PFA (in 1x PBS) for 30 min at RT. After removal of the fixative the preparations were washed for $6 \times 15$ min with 0.3% PTX (0.3% Triton X-100 in $1\times$ PBS) at RT. Blocking of samples was performed for 15 min at RT in 0.1% PBTX (0.1% BSA in 0.3% PTX). Primary antibody was diluted in 0.1% PBTX and samples were incubated at 4°C for 12 hr on a shaker. The following primary antibodies were used: chicken anti-GFP (1:500; Abcam, Cambridge, UK) and mouse anti-neurotactin (Nrt, BP106, 1:10; DSHB; RRID:AB_528404). Samples were washed in 0.3% PTX for 1 hr and Alexa-488, 568, and 647 conjugated secondary antibodies were applied in 0.1% PBTX for 2 hr. Rhodamine-conjugated phalloidin (1:200 Sigma) was used to visualize muscles.

Preparations were mounted in Vectashield mounting media (Vector Laboratories) and imaged on an Olympus FV 1000 confocal point scanning microscope. ImageJ, Adobe Photoshop and Amira 5.4.3 software (Visage Imaging, Berlin, Germany) was used for image processing and 3D reconstructions.

## Immunohistochemistry of enhancer-Gal4 lines

2–10 days old male and female flies were incubated in fixative (4% PFA in PBS, 0.2% Triton-X 100) for 3 hr at 4°C and washed with PBST (0.2% Triton-X 100) 3 × 30 min. Brain, proboscis, and head dissections were performed in PBST. Brains were dissected and transferred to a tube with ice cold PBST. Primary antibodies were incubated for 3 days at 4°C and secondary antibodies for 2 days at 4°C.

For proboscis and head dissections flies were decapitated with a razor blade. For the proboscis dissection the part of interest was isolated. For complete head dissection only a few holes were pierced into the cuticle on the ventral side of the proboscis (26-gauge needle) to allow antibody penetration. Primary antibodies were incubated for 5 days at RT and secondary antibodies for 3 days at RT.

Antibodies were diluted in PBST and used at the following concentrations: mouse anti-Bruchpilot (nc82; RRID:AB_2314868) 1:200, mouse anti-Synapsin (3c11; RRID:AB_528479) 1:100 (both obtained from Developmental Studies Hybridoma Bank, IA), rabbit anti-Discs-large (*Pielage et al., 2011*) 1:1000, rabbit anti-GFP (A6455, Life technologies, ThermoFisher, Waltham MA) 1:1000, mouse anti-mCherry (632543, Clontech, Takara, Mountain View CA) 1:1000, phalloidin Alexa 647 (Life technologies) 1:1,000. Alexa 488, 555, and 647-coupled secondary antibodies (Life technologies) were used at 1:1000.

Brains, proboscises, and heads were mounted in Vectashield and images were acquired with a Zeiss LSM 700/710 laser scanning confocal microscope with a 10x (NA 0.3) objective, a 20x (NA 0.7) oil immersion objective, or a 40x (NA 1.25–0.75) oil immersion objective. Images were processed using Imaris (Bitplane) and Adobe Photoshop software.

## GFP reconstitution across synaptic partners (GRASP)

Enhancer-*Gal4* lines were crossed to *Gr5a-LexA; UAS-tdTomato::LexAop2-CD4-spGFP11; UAS-CD4-spGFP1–10* and offspring with the genotype *Gr5a-Lexa/+; UAS-tdTomato::LexAop-CD4-spGFP11/+; UAS-CD4-spGFP1–10/enhancer-Gal4* was dissected in ice cold PBST. Brains were incubated in fixative for 20 min at 4°C and washed with PBST 3 × 30 min. Primary and secondary antibodies were incubated overnight at 4°C.

## Analysis of fly behavior

For all behavior experiments 2–10 days old male and female flies were used. Fed or starved (24 hr) flies were mounted on a glass coverslip 30 min prior to testing. The PER was analyzed using a custom-made, temperature-controlled chamber and recorded with a Canon EOS 60D camera at 25 or 50 frames/s.

### Analysis of the PER motion pattern

PER was elicited by application of 200 mM sucrose to the anterior legs. Videos were analyzed using *Adobe Premiere Pro CC* and the initiation time point of each movement (i.e. rostrum lifting, haustellum extension, labella extension, and labella spreading) as well as the time point of sucrose stimulation was measured.

### Artificial activation using TrpA1

Enhancer-Gal4 lines were crossed to UAS-TrpA1 at 22°C. The behavior was analyzed in a custom-made heating chamber and monitored at control (22°C) and activation (28–32°C) temperature. Numbers of analyzed animals are indicated in *Supplementary file 1* as responding animals/total animals.

### Artificial activation using Chrimson

Enhancer-Gal4 lines were crossed to UAS-Chrimson at 25°C and kept in the dark. Crosses were raised on standard food mixed with 200 uM all-trans retinal. The behavior, with and without

gustatory stimulation, was analyzed and monitored under control (475 nm) and activation wavelength (633 nm) in a dark room.

## Artificial silencing using shibire[ts]

Enhancer-Gal4 lines were crossed to UAS-shibire[ts] at 22°C. To elicit PER a positive stimulus (200 mM Sucrose) was applied to the anterior legs. First, PER was observed at 22°C. Flies that showed no or only incomplete PER were excluded. Second, after the chamber was heated to 29°C flies were repeatedly stimulated to deplete the synaptic vesicle pool and PER was analyzed at the restrictive temperature. Third, after the chamber was cooled to 22°C responsiveness was tested again. All flies that showed no or only incomplete PER at this stage were excluded from the analysis.

## Quantification of proboscis displacement

Enhancer-Gal4 lines were crossed to UAS-Chrimson at 25°C and kept in the dark. Crosses were raised on standard food mixed with 200 uM all-trans retinal. The behavior was analyzed and monitored in a dark room. The maximum proboscis extension (MPE) is defined as the distance between the most anterior part of the eye and the tip of the labella when the proboscis is maximally extended. One dataset consists of four MPE data points: Two at blue light with (blue[+]) and without (blue[-]) sucrose stimulation and two at red light with (red[+]) and without stimulation (red[-]). The data points at blue light and red light for one dataset are from two consecutive stimulations. In all quantifications for *Figure 9A–E* values are normalized to (blue[+]-blue[-]) which represents 100% proboscis extension distance. The values for *Figure 9A–E* are calculated the following: $\frac{blue+\ (-)blue-}{blue+\ (-)blue-}*100\%$ for grey bars, $\frac{red-\ (-)blue-}{blue+\ (-)blue-}*100\%$ for green bars and $\frac{red+\ (-)blue-}{blue+\ (-)blue-}*100\%$ for green+blue bars. Data are presented as mean ± SEM. For *Figure 9F* the values are normalized to [(blue[+]-blue[-]) – (red[-]-blue[-])] to neglect the distance that is reached due to red light alone thereby focusing on the distance that is added upon sucrose stimulation. These values are calculated the following: $\frac{(red+\ (-)blue-)\ -\ (red-\ (-)blue-)}{(blue+\ (-)blue-)\ -\ (red-\ (-)blue-)}*100\%$.

Ten stimulations on >5 different flies were used for quantifications (except for control flies: 8 stimulations on 3 different flies). Data are presented as mean ± SEM.

## Statistical analysis

We used d'Agostino-Pearson omnibus normality test to test for Gaussian distributions.

*Figure 1*: Quantification of the initiation time points of individual steps during PER: Individual flies (flies A, B, and C) were compared to each other using the Mann-Whitney U test.

*Figure 5—figure supplement 1*: Quantification of the initiation time points of individual steps during PER: w[1118] flies were compared to GMR-Gal4>shibire[ts] flies using the Mann-Whitney U test.

*Supplementary file 1* — for *Figures 5*, *6*, *7*, *8* and *Figure 8—figure supplement 1*: Quantification of animals showing behavioral phenotypes: Experimental flies (Gal4/UAS) were compared to control flies (w[1118]; Gal4/+; UAS/+) using the Wilcoxon signed-rank test.

*Figure 9*: Proboscis displacements of the same set of flies under different conditions were compared using a paired, parametric t-test.

For all statistical tests asterisks indicate: *p<0.05; **p<0.01; ***p<0.001.

## Acknowledgements

We thank Yunpo Zhao and Dominique Siegenthaler for help during initial phases of the project and for input to the manuscript. We thank all members of the Reichert, VijayRaghavan and Pielage lab for critical discussions throughout the project.

## Additional information

### Competing interests

KV: Senior editor, *eLife*. The other authors declare that no competing interests exist.

## Funding

| Funder | Author |
|---|---|
| Schweizerischer Nationalfonds zur Förderung der Wissenschaftlichen Forschung | Heinrich Reichert Jan Pielage |
| National Centre for Biological Sciences | Ali Asgar Bohra Krishnaswamy VijayRaghavan |
| J.C. Bose Fellowship of the Department of Science and Technology and CEFIPRA | Krishnaswamy VijayRaghavan |

The funders had no role in study design, data collection and interpretation, or the decision to submit the work for publication.

## Author contributions

OS, Conceptualization, Data curation, Formal analysis, Writing—original draf, Writing—review and editing, Designed the project, Performed all main experiments and analyzed the data, Wrote the manuscript with contributions from all authors; AAB, Data curation, Formal analysis, Writing—original draft, Writing—review and editing, Generated the MARCM data for Figures 3 and 4 and analyzed the data; XL, Data curation, Formal analysis, Writing—original draft, Writing—review and editing, Generated the MARCM clone for Figure 2E and contributed to the behavioral activation screen data; HR, Conceptualization, Formal analysis, Supervision, Funding acquisition, Writing—original draft, Writing—review and editing, Designed the project, Generated the MARCM data for Figures 3 and 4 and analyzed the data, Wrote the manuscript with contributions from all authors; KV, Conceptualization, Formal analysis, Supervision, Funding acquisition, Writing—original draft, Writing—review and editing, Designed the project, Generated the MARCM clone for Figure 2E and contributed to the behavioral activation screen data; JP, Conceptualization, Formal analysis, Supervision, Funding acquisition, Writing—original draft, Project administration, Writing—review and editing, Designed the project, Performed all main experiments and analyzed the data, Wrote the manuscript with contributions from all authors

## Author ORCIDs

Krishnaswamy VijayRaghavan, http://orcid.org/0000-0002-4705-5629

Jan Pielage, http://orcid.org/0000-0002-5115-5884

# Additional files

## Supplementary files

• Supplementary file 1. Quantification and statistical analysis of behavioral phenotypes. Numbers are shown as flies displaying phenotype/total flies analyzed.Experimental flies (Gal4/UAS) were compared to control flies ($w^{1118}$; Gal4/+; UAS/+) using a Wilcoxon signed-rank test.* GMR26A01, cha-Gal80>Chrimson flies were starved for 24 hr prior to testing.

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
