## [Decision Letter]

Thank you for submitting your article "Motor control of *Drosophila* feeding behavior" for consideration by *eLife*. Your article has been reviewed by three peer reviewers, one of whom, Jan-Marino Ramirez (Reviewer #1), is a member of our Board of Reviewing Editors, and the evaluation has been overseen by Eve Marder as the Senior Editor. One other individual involved in review of your submission has agreed to reveal his identity: Benjamin Eaton (Reviewer #3).

The reviewers have discussed the reviews with one another and the Reviewing Editor has drafted this decision to help you prepare a revised submission.

Summary:

The manuscript by Schwartz et al., "Motor control of *Drosophila* feeding behavior" will have a great impact for understanding the control of stereotypic motor sequences, which is of great interest for a general readership. The study provides a first detailed description of most of the motor neurons and muscles involved in feeding motor program in *Drosophila*, focusing on the movement of proboscis. The authors provide a detailed description of the proboscis muscles and an in depth anatomical analysis of the motor neurons. Using a combination of MARCM with backfilling of the proboscis nerves with rhodamin – labeled dextran dye they indentify ~20 distinct motor neurons in each hemisphere that control the activity of 13 muscle groups involved in proboscis movement and potentially in pharyngeal pumping. To functionally characterize motor neurons involved in different phases of proboscis extension the authors carried out a Gal4 / UAS activity manipulation screen using dTRPA1, Chrimson and shibirets, looking for either heat induced and optogenetic proboscis extension or a specific deficit in proboscis extension triggered by the application of sweet solution to the tarsi or by optogenetic activation of Gr5a expressing sweet sensing neurons. As a result the authors were able to identify motor neurons controlling the lifting of rostrum (MN9; also previously characterized by Gordon et al., 2009); flipping down of the haustellum – MN2; extending of the labella MN6, spreading the labella MN8 and for proboscis retraction – MN1. The manuscript is well written and is easy to follow. The results look solid (especially when looking at the videos that the authors provide), but some modifications need to be done before this manuscript can be accepted:

Essential revisions:

1) The reviewers concluded that the characterization of the behavioral phenotype is a major weakness of this study and the analysis should be in line with the previous study investigating the circuitry controlling sequential motor programs involved in grooming (Seeds et al., 2014 *eLife*). Hence, the authors need to improve the analysis of the behavioral data. Currently throughout the manuscript there is hardly any use or mentioning of control experiments. It is not clear at this stage how often genetic control animals show the described phenotypes. We assume that the reason for this is that these phenotypes are rarely/never observed in control animals. But having a solid description of the controls is absolutely required. This refers all of the behavioural experiments reported in the manuscript. So the questions are: How do control animals respond to light / heat stimuli? What is the likelihood that the phenotypes the authors observe occur in the control animals? The authors need to provide more explicit statistical measures of the strength of the phenotypes and compare them to control animals. This is for example important in the experiment described in subsection “MNs controlling haustellum extension”. In general, the authors could present stronger quantitative evidence that the proboscis extension involves a strict temporally ordered series of events. It is for example not clear from the data how consistent the series of events truly is from one animal to the next, or even from one PER to the next. Could the authors provide some metrics to support that the extension of the proboscis occurs exactly via the same series of events in their PER paradigm? What about a top down view of the PER to investigate left vs right bias? I believe this rigor is critical to ruling out that PER involves some version the chain reflex sequence.

2) The reviewers also felt that the manuscript would significantly profit from an neuroanatomical analysis of the spatial relationship of all the motor neurons in the SEZ. The current description is highly anecdotal. This can be easily done using 3D registration or co-labeling experiments.

3) The reviewers would also like to see a broader discussion of the control of movements, not only in invertebrates but also mammalian systems. Much is known about the control of arm reaching movements. It would be interesting to draw conclusions – similarities and differences. We understand that the level of understanding is different, but the question how such movements are controlled is a general one and has captivated not only those interested in invertebrate systems.

4) Another weakness of the study relates to the notion that sensory feedback does not play a role during proboscis extension. The nan mutant analysis is compelling but there are other mutations that affect proprioception in *Drosophila* which might be more applicable to control of proboscis extension (ppk, trp-γ, stum) and would increase the rigor of this analyses. In addition to proprioception, sensory feedback could manifest in numerous ways. For PER, it seems possible (even likely) that starvation (vs satiation) could change the drive to this motor behavior. Does starved vs satiated make any differences in the effects of the MN silencing on proboscis extension? For this analysis, a quantitative measure of the series of events would be helpful. At least the authors need to soften their language when stating that sensory feedback can be excluded.

[Editors' note: further revisions were requested prior to acceptance, as described below.]

Thank you for resubmitting your article "Motor control of *Drosophila* feeding behavior" for consideration by *eLife*. Your article has been reviewed by three peer reviewers, one of whom, Jan-Marino Ramirez (Reviewer #1), is a member of our Board of Reviewing Editors, and the evaluation has been overseen by Eve Marder as the Senior Editor. The following individual involved in review of your submission has agreed to reveal his identity: Benjamin Eaton (Reviewer #3).

The reviewers have discussed the reviews with one another and the Reviewing Editor has drafted this decision to help you prepare a revised submission.

Summary:

The authors have addressed the majority of our comments/concerns. The result is an interesting manuscript that uses the *Drosophila* proboscis extension as a model to understand the control of a reaching behaviour. The focus on the careful characterisation of the action patterns of individual motoneurons makes this a very comprehensive study of general interest. We have only one minor comments that can be addressed in the Discussion section.

Essential revisions:

While we agree that the data clearly support the authors' conclusion that during the physical extension of the proboscis the MNs act independently. However, we have a cautionary note regarding two of the authors' statements:

1) "These data indicate that serial execution of the individual movements is necessary to achieve the temporal precision observed in wild type flies."

2) "However, the selective block of individual MNs resulted in perturbations of the stereotypic temporal profile of the PER motion."

While we agree that these observations could be the result of "anatomic constraints" or "sensory feedback" we are not convinced that the authors can exclude the possibility that the serial execution of the motor sequence is somehow involved in the precise temporal control of this movement. We therefore encourage the authors to include this possibility in the Discussion.

---

## [Author Response]

*Essential revisions:*

*1) The reviewers concluded that the characterization of the behavioral phenotype is a major weakness of this study and the analysis should be in line with the previous study investigating the circuitry controlling sequential motor programs involved in grooming (Seeds et al., 2014 eLife). Hence, the authors need to improve the analysis of the behavioral data.*

In general, the authors could present stronger quantitative evidence that the proboscis extension involves a strict temporally ordered series of events. It is for example not clear from the data how consistent the series of events truly is from one animal to the next, or even from one PER to the next. Could the authors provide some metrics to support that the extension of the proboscis occurs exactly via the same series of events in their PER paradigm?

We address these points by providing a large new dataset showing a detailed characterization of the PER motor sequence in wild type animals. In the new Figure 1, the new Video 1 and the new Figure 5—figure supplement 1 we analyze and compare the PER sequence at high resolution using similar approaches as presented in Seeds et al., 2014.

Video 1 shows the PER motor sequence in high resolution in real time and in slow motion enabling direct visualization of the sequential progression of the individual movements. We quantified this behavior both for multiple stimulations of individual flies and across 12 different animals. Our data demonstrates that flies strictly follow the proposed sequence with only 4 out of 93 PER events displaying a small deviation where labellum extension preceded haustellum extension. In addition, these data enabled a precise quantification of the time periods between the initiation of the individual proboscis movements. Our data shows a rapid progression of the individual movements when we set initiation of rostrum lifting as time zero (Figure 1 and Figure 5—figure supplement). Between individual flies we frequently observe small but significant differences for the time periods from stimulation and rostrum lifting and from labella extension to labella spreading. Both these points are likely due to technical limitations of our assay. The stimulation of forelegs using a sucrose-soaked tissue likely induces variances in the stimulation strength of gustatory receptor neurons. Similarly, under natural conditions initiation of the spreading of the labella likely involves mechanical stimulations of the sensory neurons at the tip of the labella. However, only leg stimulations allow us to perform a selective stimulation without perturbing the motor sequence. Consistent with these observations we did not observe differences in the temporal profile of fed flies but these flies often failed to spread the labella.

Together these data now provide a qualitative and quantitative basis for all subsequent manipulations of the motor sequence (e.g. analysis after MN silencing – new Figure 5—figure supplement 1).

*Hence, the authors need to improve the analysis of the behavioral data. Currently throughout the manuscript there is hardly any use or mentioning of control experiments. It is not clear at this stage how often genetic control animals show the described phenotypes. We assume that the reason for this is that these phenotypes are rarely/never observed in control animals. But having a solid description of the controls is absolutely required. This refers all of the behavioural experiments reported in the manuscript. So the questions are: How do control animals respond to light / heat stimuli? What is the likelihood that the phenotypes the authors observe occur in the control animals?*

We are sorry for the omission of detailed control data in the original submission. Indeed, we never observed any of the described phenotypes in control animals. We are now providing extensive control data sets (*w1118*, Gal4/+, UAS/+, TrpA1/+, Chrimson/+, shi^ts^/+) for all behavior data sets. In addition, we increased the number of experimental animals for all datasets and provide appropriate statistical comparisons. The new data is presented in: Figure 5, Figure 6, Figure 7, Figure 8, Figure 8—figure supplement 1 and [Supplementary-material SD1-data].

*This is for example important in the experiment described in subsection “MNs controlling haustellum extension”.*

We now carefully evaluated the experiment originally described in subsection “MNs controlling haustellum extension”. In this experiment we use cha-Gal80 to restrict the expression of line GMR26A01 to MN2. TrpA1 activation of GMR26A01, cha-Gal80 flies is no longer sufficient to induce haustellum extension. A likely reason for this failure is low Gal4 expression levels in MN2 due to leakiness of the cha-Gal80 line. However, using Chrimson (the stronger and more precise activator) we were able to induce haustellum extension in 4 out of 67 tested animals ([Supplementary-material SD1-data]). Interestingly, in two of these flies we observed an unusual unilateral extension of the haustellum to one side of the fly. We then directly monitored Chrimson expression in one of these flies (Chrimson-mVenus) and could demonstrate that Chrimson is only expressed in a unilateral MN2 correlating with the observed behavior. These data are presented in the new Figure 6—figure supplement 1 and Video 3 and indicate that MN2 controls extension of the haustellum.

*What about a top down view of the PER to investigate left vs right bias?*

As requested and further motivated by our results for MN2 we now show a top down view of the PER with a selective stimulation of the right front leg in Video 12. Flies can perfectly discriminate between left and right stimuli and direct the proboscis towards the food source. This data is consistent with prior observations in the blow fly. Together with our neuroanatomical data these results indicate that the specific dendritic organization and the strict ipsilateral muscle innervation of proboscis movement MNs enable a precise translation of a unilateral stimulus into a directed motor behavior.

*2) The reviewers also felt that the manuscript would significantly profit from an neuroanatomical analysis of the spatial relationship of all the motor neurons in the SEZ. The current description is highly anecdotal. This can be easily done using 3D registration or co-labeling experiments.*

We now provide two new figures addressing the spatial relationship of motoneurons in the SEZ. In the new Figure 10—figure supplement 1 we provide frontal, top and side views of individual proboscis movement MNs and of GR5a-sensory axonal projections to highlight the unique anteroventral positioning of MN dendrites. This comparison clearly shows the spatial distance between axon terminals of line GR5a and dendrites of the proboscis MNs and provides support for the GRASP analysis presented in Figure 10—figure supplement 2.

In addition, we tested potential LexA derivatives of our MN-Gal4 lines for co-expression with the Gal4 lines to perform co-labelling experiments. GMR81B12-LexA is only expressed in MN6 (and one additional bilateral pair of neurons) and the expression pattern completely overlapps with GMR81B12-Gal4 (Figure 11). While expression levels of the GMR81B12-LexA line are significantly weaker compared to the Gal4 line we were able to use the line to co-label MN6 and either MN2 (GMR26A01, cha-Gal80) or MNs 1, 4, 7, 8 (GMR58H01). Both in maximum projections and in single sections the close spatial relationship between these MN dendrites is evident. This data is presented in the new Figure 11.

*3) The reviewers would also like to see a broader discussion of the control of movements, not only in invertebrates but also mammalian systems. Much is known about the control of arm reaching movements. It would be interesting to draw conclusions – similarities and differences. We understand that the level of understanding is different, but the question how such movements are controlled is a general one and has captivated not only those interested in invertebrate systems.*

We agree that a comparative discussion of the control of directed limb movements in invertebrates and vertebrates would be of general interest. However, we also strongly think that our current manuscript would not be the appropriate place for such a discussion. In the fly system we still have only very limited knowledge regarding the upstream circuit organization of the proboscis motor system. Indeed, thus far not a single interneuron has been described that is directly connected to proboscis MNs. While our study provides the basis for such studies we think that at the current stage a comparison to voluntary limb movements like arm reaching in mammalian systems would be better suited for a review. We hope the reviewers agree with us on this point.

*4) Another weakness of the study relates to the notion that sensory feedback does not play a role during proboscis extension. The nan mutant analysis is compelling but there are other mutations that affect proprioception in Drosophila which might be more applicable to control of proboscis extension (ppk, trp-γ, stum) and would increase the rigor of this analyses. In addition to proprioception, sensory feedback could manifest in numerous ways. For PER, it seems possible (even likely) that starvation (vs satiation) could change the drive to this motor behavior. Does starved vs satiated make any differences in the effects of the MN silencing on proboscis extension? For this analysis, a quantitative measure of the series of events would be helpful. At least the authors need to soften their language when stating that sensory feedback can be excluded.*

We now tested additional mutations affecting proprioception (*trp-γ, stum*) and did not observe obvious alterations of the wild type PER sequence. However, the quantification of the temporal PER sequences after silencing of individual MNs presented in the new Figure 5—figure supplement 1 showed clear deviations from the wild type profile. We thus cannot exclude the possibility that sensory feedback mechanisms may contribute to the robustness of proboscis motion. For these reasons we removed the preliminary analysis of potential proprioceptive mechanisms from the manuscript.

[Editors' note: further revisions were requested prior to acceptance, as described below.]

*Essential revisions:*

*While we agree that the data clearly support the authors' conclusion that during the physical extension of the proboscis the MNs act independently. However, we have a cautionary note regarding two of the authors’ statements:*

*1) "These data indicate that serial execution of the individual movements is necessary to achieve the temporal precision observed in wild type flies."*

*2) "However, the selective block of individual MNs resulted in perturbations of the stereotypic temporal profile of the PER motion."*

*While we agree that these observations could be the result of "anatomic constraints" or "sensory feedback" we are not convinced that the authors can exclude the possibility that the serial execution of the motor sequence is somehow involved in the precise temporal control of this movement. We therefore encourage the authors to include this possibility in the Discussion.*

Many thanks for pointing out that we didn´t include alternative possibilities in the corresponding Discussion section. We have now changed the relevant section in the Discussion.

“In addition, we provide evidence that initiation of individual movements does not depend on the execution of earlier steps of the motor sequence. However, a serial execution of the individual movements is necessary to achieve temporal precision of the PER sequence as inhibition of individual MNs led to perturbations of the stereotypic temporal profile of the PER motion. Three reasons may account for these alterations: First, the central control of serial activation of individual MNs may be perturbed by the artificial silencing of one of the target MNs. Second, the failure to move parts of the proboscis may affect sequence execution due to anatomical constraints. Third, altered sequence execution may affect sensory feedback systems as observed in the leg motor system of *Drosophila* (Mendes et al., 2013).”